eLife | RESEARCH ARTICLE

# An autoinhibitory clamp of actin assembly constrains and directs synaptic endocytosis

Steven J Del Signore[1]*, Charlotte F Kelley[1], Emily M Messelaar[1], Tania Lemos[1], Michelle F Marchan[1], Biljana Ermanoska[1], Markus Mund[2], Thomas G Fai[3], Marko Kaksonen[2], Avital Adah Rodal[1]*

[1]Department of Biology, Brandeis University, Walltham, United States; [2]Department of Biochemistry and NCCR Chemical Biology, University of Geneva, Geneva, Switzerland; [3]Department of Mathematics, Brandeis University, Waltham, United States

**Abstract** Synaptic membrane-remodeling events such as endocytosis require force-generating actin assembly. The endocytic machinery that regulates these actin and membrane dynamics localizes at high concentrations to large areas of the presynaptic membrane, but actin assembly and productive endocytosis are far more restricted in space and time. Here we describe a mechanism whereby autoinhibition clamps the presynaptic endocytic machinery to limit actin assembly to discrete functional events. We found that collective interactions between the *Drosophila* endocytic proteins Nwk/FCHSD2, Dap160/intersectin, and WASp relieve Nwk autoinhibition and promote robust membrane-coupled actin assembly in vitro. Using automated particle tracking to quantify synaptic actin dynamics in vivo, we discovered that Nwk-Dap160 interactions constrain spurious assembly of WASp-dependent actin structures. These interactions also promote synaptic endocytosis, suggesting that autoinhibition both clamps and primes the synaptic endocytic machinery, thereby constraining actin assembly to drive productive membrane remodeling in response to physiological cues.

*For correspondence:
sdelsig@gmail.com (SJDS);
arodal@brandeis.edu (AAR)

**Competing interests:** The authors declare that no competing interests exist.

## Introduction

At neuronal presynaptic terminals, actin assembly affects many physiological processes including synapse morphogenesis, traffic of numerous vesicular cargoes, and synaptic vesicle endocytosis, organization, and mobility (*Dillon and Goda, 2005*; *Nelson et al., 2013*; *Papandréou and Leterrier, 2018*). However, the molecular mechanisms that control F-actin dynamics in space and time at presynaptic membranes are largely unknown. Presynaptic terminals maintain constitutively high local concentrations of actin-associated endocytic regulatory proteins at synaptic membranes (*Reshetniak et al., 2020*; *Wilhelm et al., 2014*), yet only a small fraction of this protein pool is likely to be active at any point in time (in response to vesicle release) and space (at <100-nm-diameter endocytic sites), suggesting that the endocytic machinery is held in an inactive state at synaptic membranes. However, we do not know the mechanisms that maintain this machinery in an inactive state at the membrane, or how it is activated when and where it is needed.

One plausible mechanism to restrict membrane-cytoskeleton remodeling and endocytic activity to specific locations and times may lie in autoinhibition, which is a property of multiple endocytic proteins (*Gerth et al., 2017*; *Kim et al., 2000*; *Rao et al., 2010*; *Stanishneva-Konovalova et al., 2016*). One example is the F-BAR-SH3 protein Nervous Wreck (Nwk), which regulates synaptic membrane traffic at the *Drosophila* neuromuscular junction (NMJ) (*Coyle et al., 2004*; *O'Connor-Giles et al., 2008*; *Rodal et al., 2008*; *Rodal et al., 2011*; *Ukken et al., 2016*) and whose

**eLife digest** Neurons constantly talk to each other by sending chemical signals across the tiny gap, or 'synapse', that separates two cells. While inside the emitting cell, these molecules are safely packaged into small, membrane-bound vessels. Upon the right signal, the vesicles fuse with the external membrane of the neuron and spill their contents outside, for the receiving cell to take up and decode.

The emitting cell must then replenish its vesicle supply at the synapse through a recycling mechanism known as endocytosis. To do so, it uses dynamically assembling rod-like 'actin' filaments, which work in concert with many other proteins to pull in patches of membrane as new vesicles. The proteins that control endocytosis and actin assembly abound at neuronal synapses, and, when mutated, are linked to many neurological diseases. Unlike other cell types, neurons appear to 'pre-deploy' these actin-assembly proteins to synaptic membranes, but to keep them inactive under normal conditions. How neurons control the way this machinery is recruited and activated remains unknown.

To investigate this question, Del Signore et al. conducted two sets of studies. First, they exposed actin to several different purified proteins in initial 'test tube' experiments. This revealed that, depending on the conditions, a group of endocytosis proteins could prevent or promote actin assembly: assembly occurred only if the proteins were associated with membranes. Next, Del Signore et al. mutated these proteins in fruit fly larvae, and performed live cell microscopy to determine their impact on actin assembly and endocytosis.

Consistent with the test tube findings, endocytosis mutants had more actin assembly overall, implying that the proteins were required to prevent random actin assembly. However, the same mutants had reduced levels of endocytosis, suggesting that the proteins were also necessary for productive actin assembly. Together, these experiments suggest that, much like a mousetrap holds itself poised ready to spring, some endocytic proteins play a dual role to restrain actin assembly when and where it is not needed, and to promote it at sites of endocytosis.

These results shed new light on how neurons might build and maintain effective, working synapses. Del Signore et al. hope that this knowledge may help to better understand and combat neurological diseases, such as Alzheimer's, which are linked to impaired membrane traffic and cell signalling.

mammalian homolog FCHSD2 regulates endocytosis and endocytic traffic in mammalian cells (*Almeida-Souza et al., 2018*; *Xiao and Schmid, 2020*; *Xiao et al., 2018*). Nwk/FCHSD2 proteins couple two activities: membrane remodeling and WASp-dependent actin polymerization (*Almeida-Souza et al., 2018*; *Rodal et al., 2008*; *Stanishneva-Konovalova et al., 2016*). Intramolecular auto-inhibitory interactions between the Nwk F-BAR and its two SH3 domains mutually inhibit both Nwk membrane binding and activation of WASp (*Stanishneva-Konovalova et al., 2016*). Unlike other F-BAR-SH3 proteins, which are completely released from autoinhibition upon membrane binding (*Guerrier et al., 2009*; *Meinecke et al., 2013*; *Rao et al., 2010*), the SH3b domain of Nwk continues to restrict SH3a-mediated WASp activation even after Nwk binds membranes (*Stanishneva-Konovalova et al., 2016*). This suggests that autoinhibition allows Nwk-WASp to remain inactive even after recruitment to the membrane, thus keeping the endocytic machinery in a primed but inactive state. We hypothesized that additional binding partners of Nwk$^{SH3b}$ may be required to fully activate membrane remodeling at discrete times and locations at the synapse.

An excellent candidate for release of Nwk autoinhibition at synapses is the endocytic adaptor intersectin (Dap160 in *Drosophila*). Intersectin interacts with numerous endocytic proteins to regulate endocytosis in mammalian cells (*Henne et al., 2010*; *Okamoto et al., 1999*; *Praefcke et al., 2004*; *Pucharcos et al., 2000*; *Schmid et al., 2006*; *Sengar et al., 1999*; *Teckchandani et al., 2012*) and has been implicated in several steps of the synaptic vesicle cycle (*Evergren et al., 2007*; *Gerth et al., 2017*; *Jäpel et al., 2020*; *Pechstein et al., 2010*; *Pechstein et al., 2015*). Of particular note, intersectin recruits the Nwk homolog FCHSD2 to sites of endocytosis (*Almeida-Souza et al., 2018*), though it is not yet known how this affects FCHSD2 autoinhibition. In *Drosophila*, Dap160 interacts with WASp, Nwk, and other membrane-remodeling proteins via its four SH3 domains

(SH3AD), and regulates the levels and localization of many of these proteins, including Nwk (*Koh et al., 2004*; *Marie et al., 2004*; *Roos and Kelly, 1998*). Further, *dap160* mutant phenotypes overlap with those of Nwk and WASp mutants, including impaired synaptic vesicle cycling and synaptic overgrowth (*Coyle et al., 2004*; *Khuong et al., 2010*; *Koh et al., 2004*; *Marie et al., 2004*; *Winther et al., 2013*). Finally, intersectin and Dap160 shift localization from synaptic vesicle pools to the plasma membrane in response to synaptic activity (*Evergren et al., 2007*; *Gerth et al., 2017*; *Winther et al., 2015*), suggesting that Dap160 may provide the spatiotemporal link between salient physiological triggers and Nwk/WASp activation.

The high concentration and broad membrane distribution of inactive endocytic proteins (*Reshetniak et al., 2020*; *Wilhelm et al., 2014*) make it difficult to characterize the molecular dynamics of synaptic endocytosis (in contrast to non-neuronal cells; *Kaksonen and Roux, 2018*). To overcome this barrier, we quantified discrete actin assembly events at the *Drosophila* NMJ as a proxy for productive endocytosis, as actin assembly is both a primary target of the endocytic apparatus under investigation and is required for synaptic vesicle endocytosis in all forms, including at the *Drosophila* NMJ (*Kononenko et al., 2014*; *Wang et al., 2010*; *Wu et al., 2016*). This synapse is an ideal system to investigate the molecular dynamics of the endocytic machinery due to its large size, ease of genetic manipulation, and accessibility to live and super-resolution imaging. Here we combine in vitro biochemical approaches with quantitative imaging at the NMJ to define the interactions among Dap160, Nwk, and WASp that relieve autoinhibition. These interactions drive robust membrane-associated actin assembly in vitro, regulate the frequency and dynamics of synaptic actin structures in vivo, and are functionally required for normal endocytosis at the NMJ.

## Results

### Actin assembles in discrete dynamic patches despite broad distribution of presynaptic membrane-cytoskeleton-remodeling machinery

While the importance of actin in synaptic endocytosis is clear (*Kononenko et al., 2014*; *Wang et al., 2010*; *Wu et al., 2016*), until now there has been no quantitative analysis of individual actin-dependent membrane-remodeling events at synapses. To better understand presynaptic F-actin dynamics and to identify sites where the cytoskeleton- and membrane-remodeling machinery is active, we quantified individual F-actin assembly events by spinning disc confocal microscopy of NMJs presynaptically expressing fluorescent actin probes. To control for developmental variation, all experiments were performed on late third-instar larvae (~96–120 hr after egg laying) on muscle 6/7 NMJs at abdominal segments 3–4, since the development and physiology of these synapses are well characterized (*Harris and Littleton, 2015*). To control for variation in size between neurons, we normalized patch frequencies by the synapse area measured and presented data per 10 μm², which is approximately the size of a synaptic bouton in this system. We performed these experiments under resting conditions, where vesicle release is spontaneous at a rate of ~5–6 vesicles/10 μm²/min (*Akbergenova et al., 2018*; *Melom et al., 2013*), presumably requiring a similar rate of compensatory endocytosis (*Sabeva et al., 2017*).

We first compared the dynamics of three actin markers: GFP::actin, GFP-tagged moesin F-actin-binding domain (GMA), and Lifeact:: Ruby. The predominant structures labeled by these markers were transient patches at the presynaptic membrane (*Video 1*, *Figure 1A*, *Figure 1—figure supplement 1A*), as has been previously observed (*Nunes et al., 2006*; *Pawson et al., 2008*; *Piccioli and Littleton, 2014*). We then quantified individual actin patch

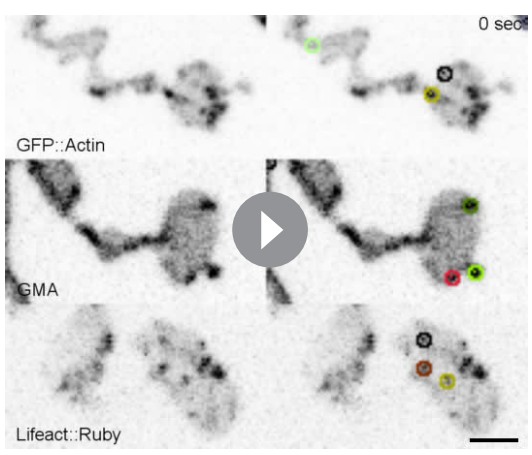

**Video 1.** Dynamics of actin patches labeled by complementary reporters.
https://elifesciences.org/articles/69597#video1

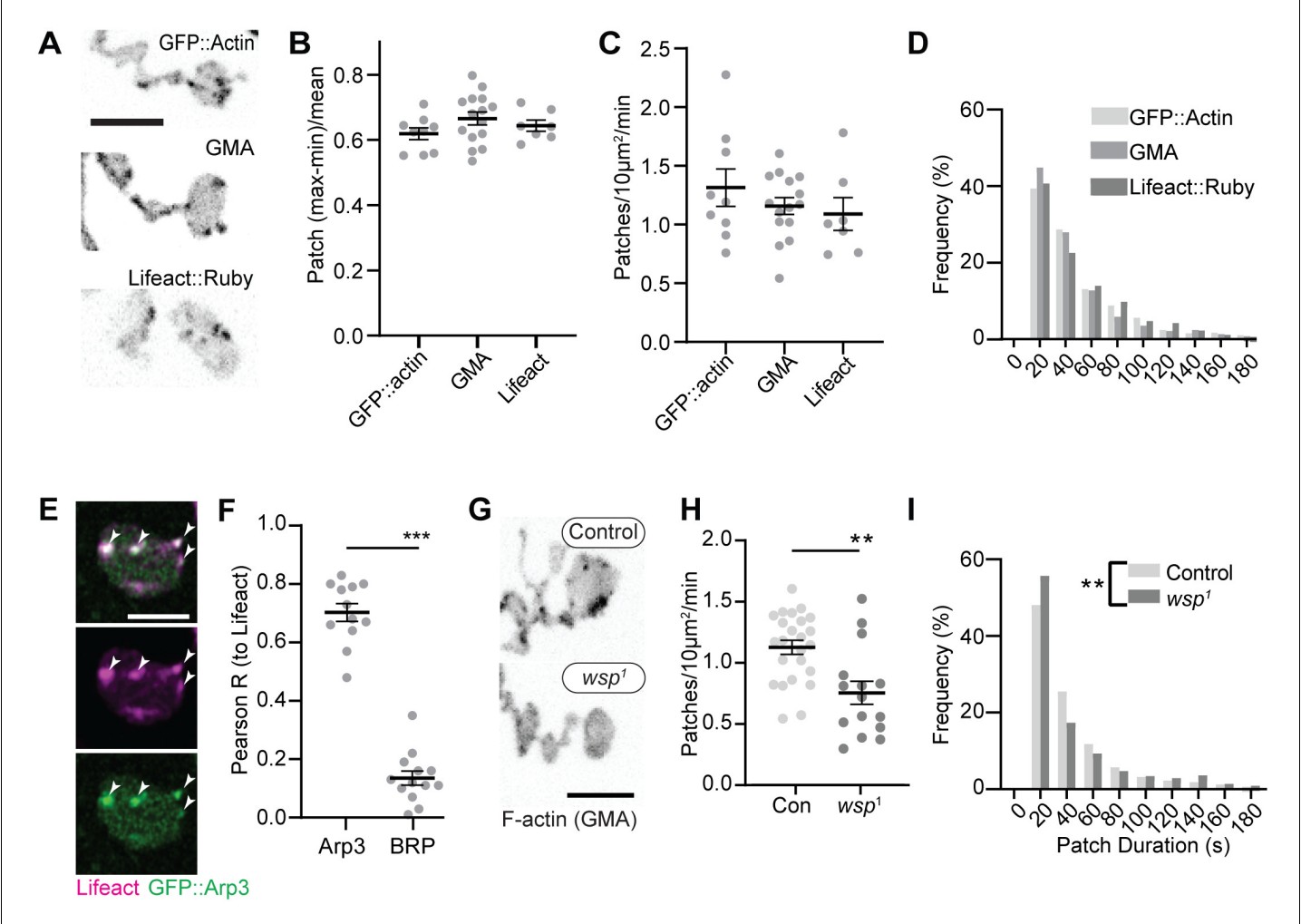

**Figure 1.** Synaptic actin patches are dynamic WASp-dependent structures. (**A**) Representative maximum intensity projections (MaxIPs) of single spinning disc confocal microscopy time points, showing C155-Gal4-driven actin probes GFP::actin, GMA, and Lifeact::Ruby. (**B–D**) Automatic detection and analysis of movies acquired at 0.25 Hz of F-actin patch intensity amplitude (**B**), frequency (**C**), and duration distribution (**D**) show similar dynamics for different reporters. (**E, F**) Single plane Airyscan image of a live muscle 6/7 neuromuscular junction (NMJ) expressing Lifeact::Ruby (magenta) and Arp3:: GFP (green). Actin patches colocalize extensively with Arp3::GFP. (**F**) Quantification of colocalization by Pearson's coefficient. Arp3 colocalizes with Lifeact significantly more than BRP::GFP, a similarly punctate and membrane-associated negative control. Graph shows mean ± sem; n represents NMJs. (**G–I**) Patch assembly requires the Arp2/3 activator WASp. GMA patch dynamics in control and WASp mutant animals imaged at 0.25 Hz. (**G**) MaxIPs of single spinning disc confocal microscopy time points, showing pan-neuronally expressed GMA localization in control and *wsp*[1] mutant muscle 6/7 NMJs. (**H**) Quantification of patch frequency. Graph shows mean ± sem; n represents NMJs. (**I**) Quantification of patch-duration distribution. Bins are 20 s; X-axis values represent bin centers. n represents patches. Scale bars in (**A**) and (**G**) are 5 μm, and scale bar in (**E**) is 2.5 μm. Associated with *Figure 1—figure supplement 1*, *Figure 1—figure supplement 2*, and *Video 1*.

The online version of this article includes the following source data and figure supplement(s) for figure 1:

**Source data 1.** Source data for *Figure 1* and associated figure supplements.

**Figure supplement 1.** Additional characterization of actin patches.

**Figure supplement 2.** Actin dynamics analysis is robust to tracking parameters.

dynamics using automated particle tracking and quantification (*Berro and Pollard, 2014*; *Tinevez et al., 2017*), which captured on the order of 30–50% of visible actin structures (see 'Materials and methods', *Figure 1—figure supplement 2*, and *Figure 6—figure supplement 1* for more details on optimization and validation of actin particle analysis). We first imaged at 0.25 Hz and measured an average of 1.2 GMA patches/10 $\mu m^2$/min, exhibiting a mean duration of 48.0 s ± 45.6 s, with an average relative amplitude of 68 ± 32% (($I_{max}$-$I_{min}$)/$I_{mean}$) (*Figure 1B–D*).

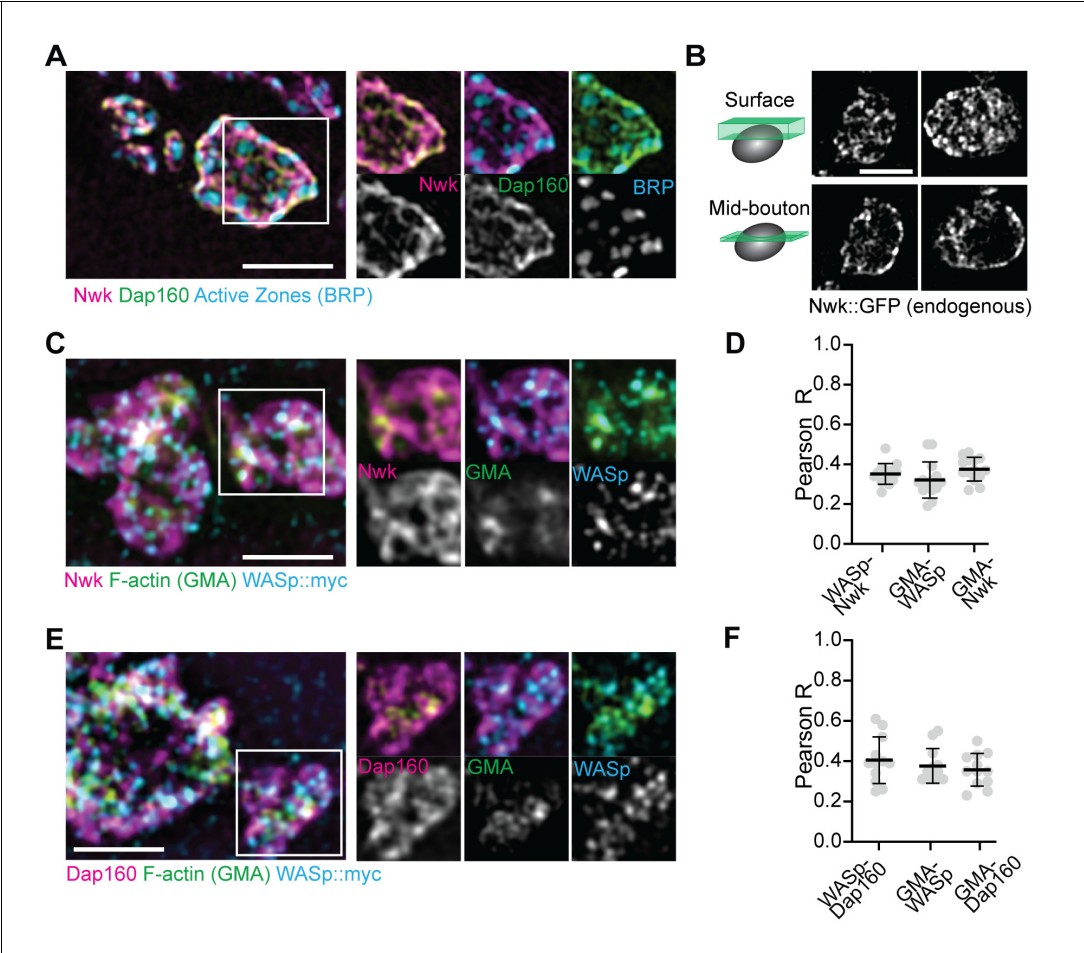

**Figure 2.** Periactive zone proteins accumulate broadly across the NMJ. (**A**) The periactive zone (PAZ) proteins Nwk (magenta) and Dap160 (green) accumulate in a micron-scale mesh adjacent to active zones (AZ) (Bruchpilot, blue). Image shows maximum intensity projection (MaxIP) of a structured illumination microscopy (SIM) Z-stack. (**B**) Surface projection (top) and medial optical section (bottom) SIM images of live-imaged endogenous Nwk:: GFP showing abundant and specific membrane recruitment, similar to fixed imaging. (**C–F**) PAZ proteins partially colocalize with actin patches. Optical slices of SIM micrographs showing F-actin (labeled with GMA) localization with presynaptically expressed WASp::Myc and Nwk (**C**) or Dap160 (**E**). (**D, F**) Quantification of colocalization between GMA and WASp::Myc, and Nwk (**D**) or Dap160 (**F**). (**D, F**) Quantification (Pearson correlation coefficient R) of colocalization between the indicated pairs of proteins. Graphs show mean ± sem; n represents neuromuscular junctions (NMJs).

The online version of this article includes the following source data for figure 2:

**Source data 1.** Source data for *Figure 2*.

Quantification of GFP::actin and Lifeact::Ruby showed very similar dynamics to GMA, suggesting that these measurements robustly reflect the underlying actin dynamics and not the specific properties of a particular probe. We did note a high percentage of patches in the minimum duration bin, suggesting the existence of even briefer patches. To address this, we also performed imaging at 1 Hz, which could not capture the entire lifetime distribution due to photobleaching but was able to identify a larger population of short-duration patches (*Figure 1—figure supplement 1B*) with an average duration of ~16 +/- 20 s. Given this range of measurements at different sampling frequencies and the efficiency of our automated detection, we estimate that patch frequency is between 2.8 and 10.3 events/10 $\mu m^2$/min (see 'Materials and methods' for calculations), on par with the expected frequency of endocytic events, and with a similar albeit broader distribution of durations compared to yeast (15 s; *Berro and Pollard, 2014*) and mammalian cells (~40 s; *Taylor et al., 2011*).

We next examined the molecular determinants of synaptic actin patch assembly. Patches strongly co-labeled with Arp3::GFP (Pearson coefficient 0.70), significantly higher than the active zone marker Bruchpilot (BRP), which served as a punctate and membrane-localized negative control (*Figure 1E–*

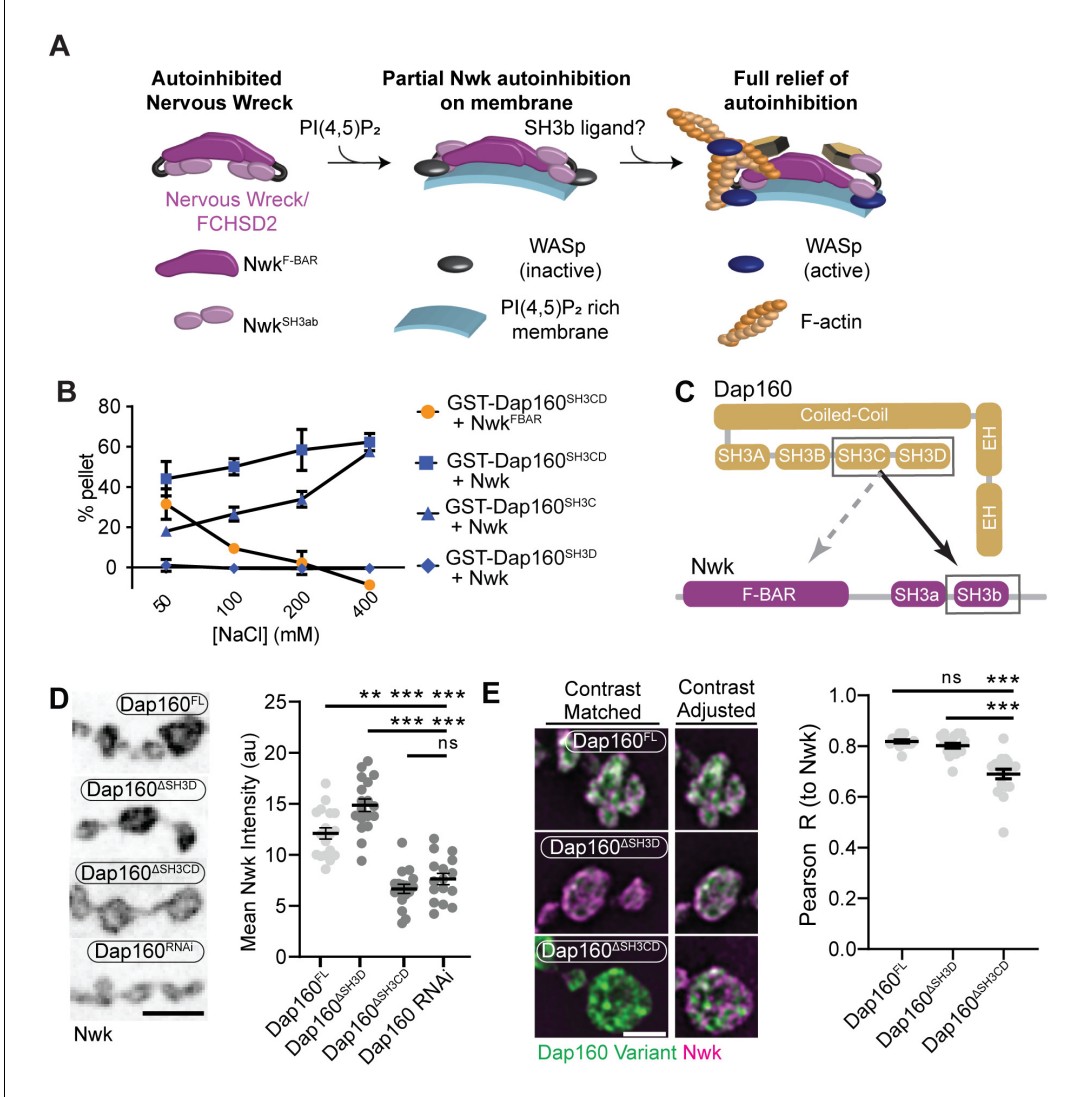

**Figure 3.** Distinct SH3-SH3 and SH3-BAR domain interactions drive Dap160-Nwk association in vitro and at synapses. (A) Model for autoinhibition of Nwk membrane binding and WASp activation. Neither membrane-bound nor membrane-free Nwk efficiently activates WASp-mediated actin polymerization, due to persistent SH3b-mediated autoinhibitory interactions, suggesting that an SH3b domain ligand is required for activation. (B) Dap160$^{SH3CD}$ exhibits electrostatic and hydrophobic interactions with the Nwk F-BAR and SH3 domains, respectively. Glutathione-S-transferase (GST) fusion proteins were immobilized on glutathione agarose and incubated with the indicated purified proteins. Pellets and supernatants were fractionated by sodium dodecyl sulphate–polyacrylamide gel electrophoresis (SDS-PAGE), Coomassie stained, and quantified by densitometry. Graphs show the average ± sem of three independent reactions. [Nwk$^{F-BAR}$] = 1.5 μM, [Nwk] = 0.8 μM, [GST-Dap160$^{SH3CD}$] = 1.6 μM, [GST-Dap160$^{SH3C/D}$] = 1.2 μM. (C) Summary of Dap160$^{SH3CD}$-Nwk$^{SH3ab}$ interactions. Gray and black arrows indicate electrostatic and hydrophobic interactions, respectively. (D, E) Maximum intensity projection (MaxIP) spinning disc confocal (D) or single Z-plane structured illumination microscopy (SIM) micrographs (E) of muscle 4 neuromuscular junctions (NMJs) expressing C155-GAL4-driven UAS-Dap160 rescue transgene variants in a *dap160* null background (dap160$^{Δ1/Df}$). Loss of the Dap160$^{SH3CD}$ domains (Dap160$^{ΔSH3CD}$), but not the SH3D domain alone (Dap160$^{ΔSH3D}$), decreases the abundance of Nwk (D, right) and Dap160-Nwk colocalization (E, right) at synapses. Contrast-matched panels in (E) are displayed with the same brightness/contrast. Adjacent panels are contrast-adjusted per image to facilitate comparison of Nwk-Dap160 colocalization. Graphs show mean ± sem; n represents NMJs. Scale bars in (D) and (E) are 5 μm and 2.5 μm, respectively. Associated with *Figure 3—figure supplements 1–2*.

The online version of this article includes the following source data and figure supplement(s) for figure 3:

**Source data 1.** Source data for *Figure 3* and associated figure supplements.

**Figure supplement 1.** Domain specific interactions between Dap160 and Nwk.

**Figure supplement 2.** Validation of Dap160 transgene rescue and knockdown experiments.

*F*). These data suggest that actin patches are predominantly composed of branched F-actin, similar to sites of endocytosis in other cell types (*Akamatsu et al., 2020*; *Collins et al., 2011*). To test whether synaptic actin patches require Arp2/3 activation, we analyzed patch dynamics in larvae lacking the Arp2/3 activator WASp. We compared a genomic mutant (*Figure 1G–I*), likely hypomorphic due to maternal contribution (*Ben-Yaacov et al., 2001*), to presynaptic depletion in neurons expressing WASp RNAi (*Figure 1—figure supplement 1 and C–F*). Using both approaches allows us to distinguish neuron-autonomous from non-autonomous effects of WASP, which is present both pre- and postsynaptically (*Coyle et al., 2004*). Both genomic and RNAi manipulations significantly reduced the number of actin patches, while genomic mutants also skewed the distribution of patch durations toward both shorter and longer events (*Figure 1I*). These differences could reflect variable loss of function between the RNAi and mutant, or identify separable presynaptic autonomous (patch frequency) vs non-autonomous (patch duration) effects of WASp. Overall, these data clearly indicate that WASp is autonomously required in neurons to initiate assembly of presynaptic actin patches, similar to its involvement in endocytosis in yeast, mammalian non-neuronal cells, and in the NMJ (*Hussain et al., 2001*; *Kessels and Qualmann, 2004*; *Khuong et al., 2010*; *Madania et al., 1999*).

We next examined the synaptic distribution of two likely WASp regulators, Nwk and Dap160. By conventional and super-resolution microscopy of neurons in diverse organisms, these and other presynaptic membrane-remodeling proteins localize to a broad membrane domain surrounding active zones, termed the periactive zone (PAZ) (*Coyle et al., 2004*; *Denker et al., 2011*; *Gerth et al., 2017*; *Koh et al., 2004*; *Marie et al., 2004*; *Sone et al., 2000*). Consistent with these prior descriptions, we observed by structured illumination microscopy (SIM) that the PAZ proteins Nwk and Dap160 localize to a membrane-proximal mesh that surrounds active zones, which were labeled with BRP (*Figure 2A*). We observed similar results by live imaging of an endogenously tagged Nwk protein by SIM, which revealed most proteins to be close to the plasma membrane (*Figure 2B*). We then compared the localization of PAZ proteins to F-actin patches at the NMJ. As expected, actin patches were much sparser than the endocytic machinery, and GMA-labeled patches only partially overlapped with each of the presynaptic WASp, Nwk, and Dap160 (*Figure 2C–F*; Pearson's coefficients of 0.38, 0.38, and 0.36, respectively). These data confirm that, in sharp contrast to the actin regulatory machinery, which localizes broadly across the PAZ, actin assembly itself is much sparser both spatially and temporally at the NMJ. This raises the question of how PAZ machinery might itself be locally regulated to promote the formation of productive synaptic actin assemblies.

## Multiple interaction interfaces between Dap160 and Nwk regulate Nwk autoinhibition

The hypothesis that PAZ protein-mediated actin assembly might be locally activated is particularly interesting given that we and others have previously shown that autoinhibition of both Nwk and its mammalian homolog FCHSD2 suppresses both WASp activation and membrane binding (see *Figure 3A* for summary model; *Almeida-Souza et al., 2018*; *Rodal et al., 2008*; *Stanishneva-Konovalova et al., 2016*). These results suggest that transient or localized relief of autoinhibition could explain how the PAZ controls actin assembly. To determine if and how the candidate activator Dap160 might relieve Nwk autoinhibition, we first mapped their specific interaction domains using glutathione-S-transferase (GST) pulldown assays and found that purified Dap160 SH3C-containing protein fragments (SH3C, SH3CD, or SH3ABCD) directly interact with Nwk$^{SH3b}$, while SH3D alone does not (*Figure 3B*, *Figure 3—figure supplement 1*; see *Figure 3—figure supplement 2A* for details of constructs used). Unexpectedly, Dap160 SH3C, SH3D, and SH3CD domain fragments also, each, interact with the isolated Nwk F-BAR domain (*Figure 3—figure supplement 1B*). We next determined how Dap160 interacts with Nwk$^{F-BAR}$ compared to a Nwk fragment containing the F-BAR and both SH3 domains. Dap160-Nwk$^{F-BAR}$ interactions were progressively eliminated by increasing salt, suggesting they are mediated by electrostatic interactions. By contrast, Dap160$^{SH3CD}$-Nwk interactions were maintained (*Figure 3B*, *Figure 3—figure supplement 1C*), suggesting that the SH3-SH3 interaction is mediated primarily by hydrophobic interactions, consistent with their mammalian homologs (*Almeida-Souza et al., 2018*; see summary of interactions in *Figure 3C*). Finally, we found that truncation of Dap160$^{SH3CD}$ decreased the levels of Nwk in synaptic boutons similarly to Dap160 knockdown (*Figure 3D*, *Figure 3—figure supplement 2B–C*). Dap160$^{ΔSH3CD}$ also exhibited reduced colocalization with Nwk compared to wild-type Dap160 (*Figure 3E*, *Figure 3—figure supplement 2C*), further supporting an in vivo requirement for this

interaction. Notably, truncation of Dap160$^{SH3D}$ did not exhibit a phenotype in these assays despite lower levels of expression (*Figure 3—figure supplement 2B*), suggesting that additional factors absent from our in vitro assays may collaborate to regulate Nwk in vivo.

## Dap160$^{SH3CD}$ and membranes relieve inhibition of Nwk-WASp-Arp2/3 actin assembly in vitro

We have previously shown that Nwk only weakly activates WASp-dependent actin assembly in vitro, due to Nwk autoinhibition (*Stanishneva-Konovalova et al., 2016*). To test whether Dap160$^{SH3CD}$

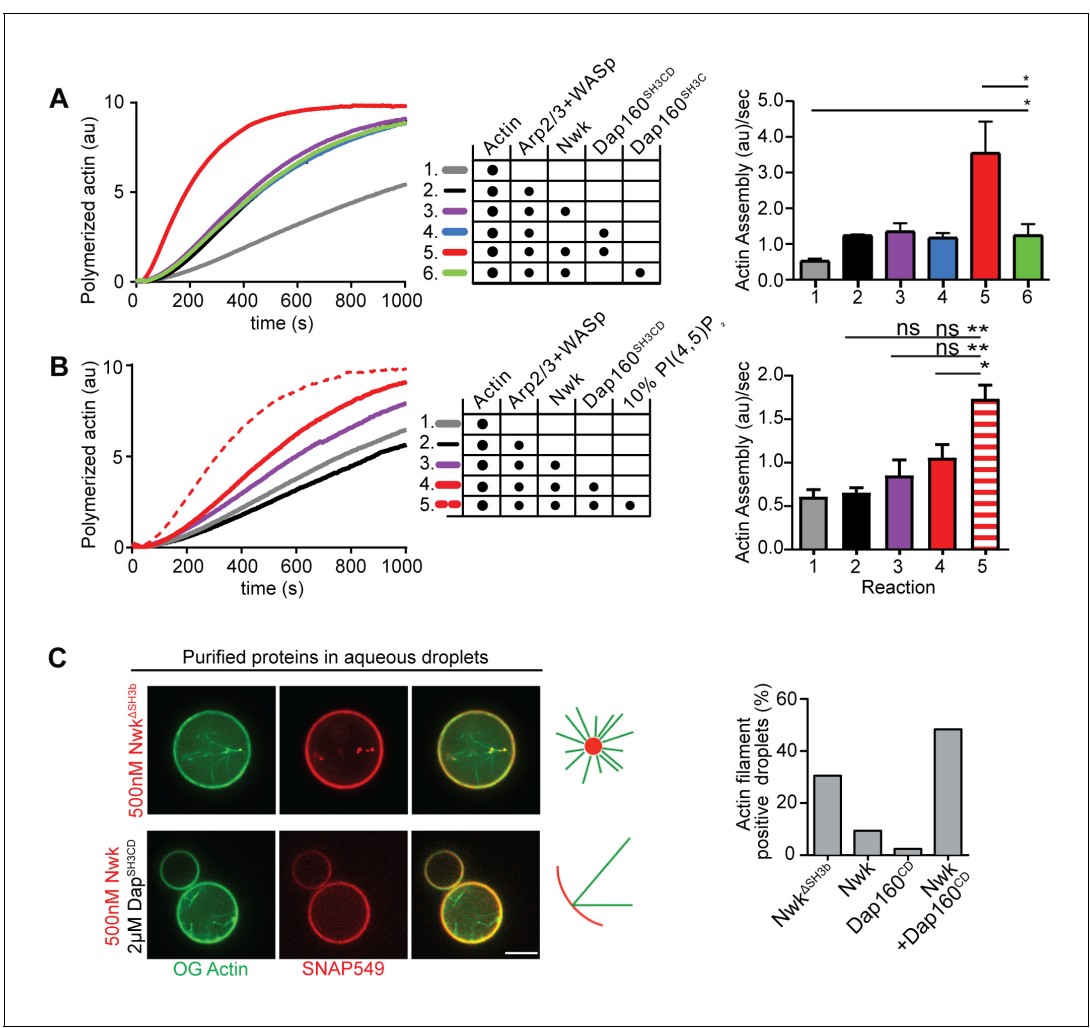

**Figure 4.** Nwk, Dap160, and PI(4,5)P$_2$ potentiate WASp-mediated actin assembly at membranes. (**A, B**) Pyrene-actin assembly assay (2.5 µM actin, 5% pyrene-labeled). Curves are representative single experiments demonstrating actin assembly kinetics; graphs represent rates calculated from the linear range of assembly from at least two independent experiments. (**A**) The combination (red trace) of Nwk and Dap160$^{SH3CD}$ enhances WASp-Arp2/3-mediated actin assembly. Either alone (magenta and blue traces) has no effect on WASp activity. [Nwk] = 500 nM, [Dap160] = 2 µM, [WASp] = 50 nM, [Arp2/3] = 50 nM. (**B**) PI(4,5)P$_2$ enhances Nwk-Dap160 activation of WASp-mediated actin assembly. Nwk alone or in combination with 10% PI(4,5)P$_2$ liposomes fails to activate WASp, while the addition of Dap160$^{SH3CD}$ and PI(4,5)P$_2$ synergistically enhances WASp-mediated actin assembly. [Nwk] = 100 nM, [Dap160] = 500 nM, [WASp] = 50 nM, [Arp2/3] = 50 nM. (**C**) Single slices from spinning disc confocal micrographs of water-droplet actin assembly assay: SNAP-labeled Nwk constructs (red) and Oregon Green actin (green) were mixed with the indicated proteins in an aqueous solution and emulsified in 97.5% 1,2-diphytanoyl-sn-glycero-3-phosphocholine (DPHPC), 2.5% PI(4,5)P$_2$ in decane. Both deregulated Nwk$^{\Delta SH3b}$ and Nwk + Dap160$^{SH3CD}$ promote F-actin assembly in droplets. However, while Nwk - Dap160$^{SH3CD}$ derived F-actin associates with the lipid interface, de-regulated Nwk$^{\Delta SH3b}$ promotes actin assembly from asters that do not associate with membranes. [Nwk$^{1-xxx}$] = 500 nM, [Dap160] = 2 µM, [WASp] = 50 nM, [Arp2/3] = 50 nM. Graph indicates percentage of droplets with observable actin filament assembly. Scale bar in (**C**) is 10 µm.

The online version of this article includes the following source data for figure 4:

**Source data 1.** Source data for *Figure 4*.

might relieve Nwk autoinhibition, we performed pyrene-actin assembly assays (*Figure 4*). At moderate Nwk-Dap160 concentrations (500 nM and 2 μm, respectively), Nwk and Dap160$^{SH3CD}$ significantly enhanced the rate of WASp-Arp2/3-mediated actin assembly compared to Nwk plus WASp alone (*Figure 4A*). This effect is through Nwk, as Dap160$^{SH3CD}$ had no effect on WASp-Arp2/3 in the absence of Nwk. Further, Dap160 enhancement of Nwk-WASp actin assembly required the Dap160$^{SH3D}$ domain, further showing that the specific Dap160$^{SH3D}$-Nwk$^{F-BAR}$ interaction relieves functional Nwk autoinhibition in vitro. Thus, multiple Nwk-Dap160 interactions work together to relieve autoinhibition of Nwk.

To generate salient physiological force, actin assembly must be coupled to membranes, and negatively charged lipids are an important ligand for both Nwk and WASp. Thus, we next tested whether addition of PI(4,5)P$_2$-rich liposomes modified actin assembly by Nwk, Dap160, and WASp (*Figure 4B*). Indeed, PI(4,5)P$_2$-containing liposomes synergistically activated WASp-mediated actin assembly in concert with Dap160 and Nwk. By contrast, neither Nwk, PI(4,5)P$_2$, nor Nwk + PI(4,5)P$_2$ on their own were sufficient to activate WASp above baseline (*Figure 4B*). Since PI(4,5)P$_2$ is also insufficient to robustly activate either WASp or Nwk under these conditions (*Stanishneva-Konovalova et al., 2016*), our data suggest that WASp activation reflects coordinated relief of Nwk autoinhibition by both Dap160 and membranes. To further explore the coupling between lipid association and actin assembly, we conducted F-actin assembly assays in a droplet assay, in which protein-containing aqueous droplets are surrounded by a lipid interface, with lipid head groups facing the aqueous phase (*Figure 4C*). In this assay, we found that coordinated interactions among Nwk, Dap160, and WASp directed actin assembly to the lipid interface. By contrast, substitution of Nwk lacking its autoinhibitory and Dap160-interacting SH3b domain (Nwk$^{ΔSH3b}$) caused actin to assemble as free-floating asters (*Figure 4C*). We have previously found that expression of a similarly deregulated fragment (Nwk$^{1-631}$) at the NMJ led to diffuse actin filament assembly throughout the synapse (*Stanishneva-Konovalova et al., 2016*). Together, these data suggest that Nwk$^{SH3b}$ has a dual role in maintaining autoinhibition via Nwk-F-BAR interactions and permitting actin assembly at specific synaptic locations via Dap160-mediated activation.

## Dap160 and WASp relieve Nwk autoinhibition and promote its membrane association

Our actin assembly data suggest that membrane recruitment is a critical regulator of the Nwk-Dap160-WASp complex (*Figure 4B–C*). To test whether Nwk-Dap160 interactions directly regulate membrane recruitment, we performed liposome cosedimentation assays. We found that Dap160$^{SH3CD}$ enhanced Nwk membrane binding in a dose-dependent fashion (*Figure 5A*). This effect depended on membrane charge, as Dap160$^{SH3CD}$ significantly enhanced Nwk membrane binding at both 5 and 10%, but not at 2.5% PI(4,5)P$_2$ (*Figure 5B*). Only at 10% PI(4,5)P$_2$ did Dap160$^{SH3CD}$ promote Nwk membrane binding to the same extent as the completely uninhibited Nwk$^{FBAR}$ domain alone, suggesting that membrane charge and intermolecular interactions with Dap160 together tune Nwk membrane recruitment. Indeed, this effect required the full Dap160$^{SH3CD}$-Nwk$^{SH3b}$ interaction: Dap160$^{SH3C}$ alone was unable to promote membrane binding by Nwk, and Dap160$^{SH3CD}$ did not enhance membrane binding of Nwk lacking its Dap160-interacting SH3b domain (*Figure 5—figure supplement 1A*). These data further support the hypothesis that Dap160$^{SH3CD}$ relieves Nwk$^{SH3b}$-mediated autoinhibition.

As we found that Dap160$^{SH3CD}$ is insufficient to fully activate membrane binding by Nwk at intermediate phosphoinositide concentrations (*Figure 5A*), we asked whether WASp could further enhance Nwk membrane recruitment. Indeed, the addition of Dap160$^{SH3CD}$ and WASp together enhanced Nwk membrane association to the level of the isolated F-BAR domain (*Figure 5C*). Moreover, coordinated binding of all three components resulted in significantly enriched membrane association of both WASp and Dap160 (*Figure 5C*). We directly observed the coordinated recruitment of Nwk and Dap160 in the presence of WASp using fluorescently labeled proteins on GUVs (*Figure 5D*). Consistent with the direct Dap160-Nwk$^{SH3b}$ interaction, we found that deletion of the Nwk$^{SH3b}$ domain abolished both the Dap160$^{SH3CD}$-dependent increase and the coordinated recruitment of WASp and Dap160 (*Figure 5—figure supplement 1A*). Notably, addition of Dap160 and WASp did not change the nature of membrane deformations generated by Nwk (scalloped and pinched membranes; *Becalska et al., 2013*), suggesting that Dap160 and WASp together potentiate

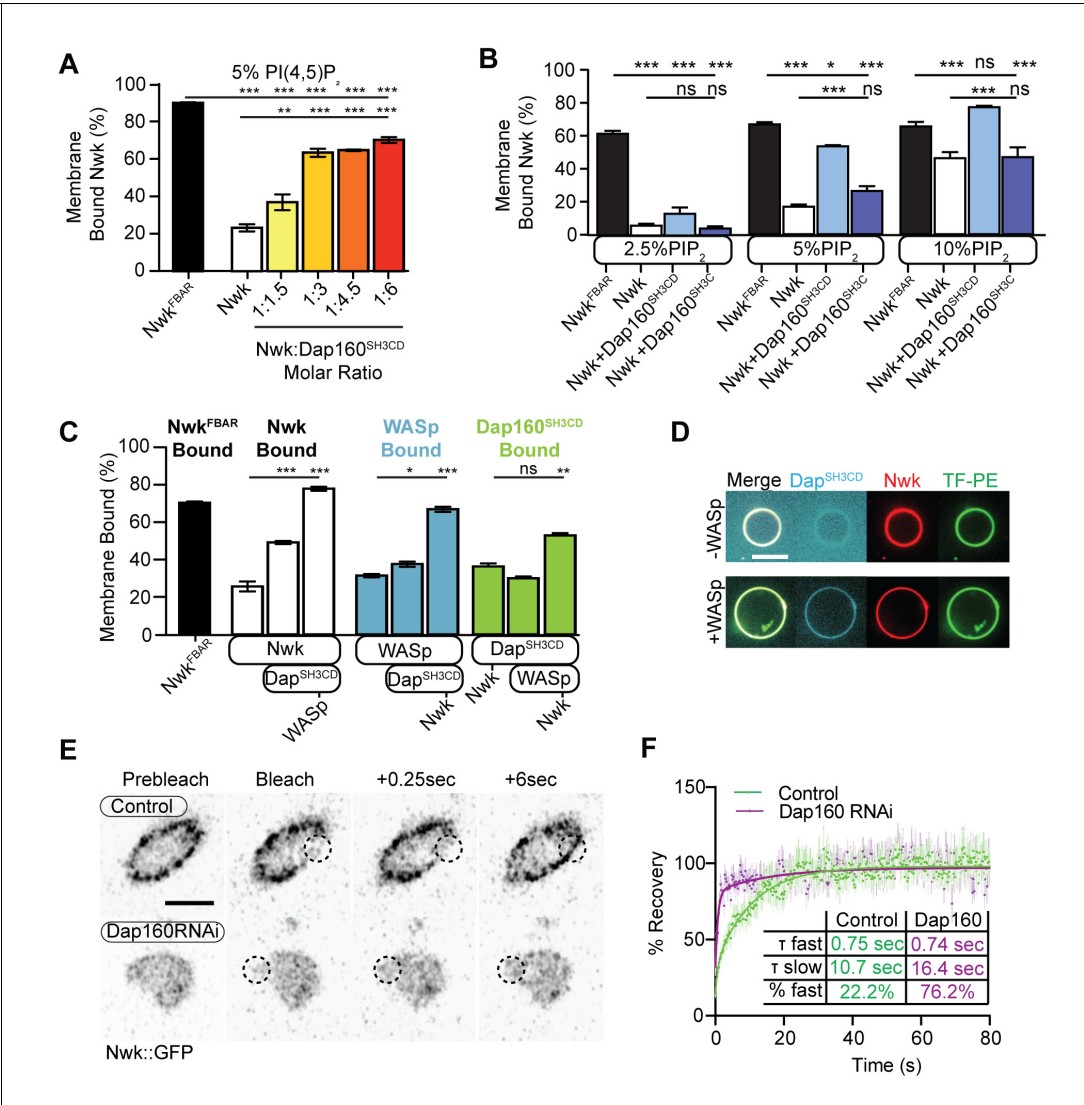

**Figure 5.** Dap160^SH3CD and WASp promote Nwk membrane association. (A–C) Liposome cosedimentation assays between the indicated purified proteins and liposomes composed of [mol% = DOPC/DOPE/DOPS/PI(4,5)P$_2$ = 80-x/15/5/x], with x representing PI(4,5)P$_2$ concentration as noted. Quantification from Coomassie-stained gels represents the mean fraction of total protein that cosedimented with the liposome pellet ± sem. (A) 1:3 Nwk:Dap160^SH3CD saturates enhancement of Nwk membrane association at 5% PI(4,5)P$_2$, but not to the level of the isolated Nwk F-BAR alone (Nwk^F-BAR, black bar). [Nwk^F-BAR] = 3 μM, [Nwk] = 1.125 μM, [Dap160^SH3CD] = 1.7–6.8 μM. (B) Dap160^SH3CD (but not Dap160^SH3C) enhances Nwk association with membranes at a range of PI(4,5)P$_2$ concentrations. Maximum binding (comparable to Nwk^F-BAR) occurs only at 5–10% PI(4,5)P$_2$ concentrations. [Nwk^1-xxx] = 2 μM, [Dap160] = 6 μM. (C) Nwk, WASp, and Dap160^SH3CD mutually enhance membrane recruitment. Addition of Dap160^SH3CD and WASp additively enhances Nwk membrane association, while Dap160^SH3CD and WASp show maximum recruitment to 10% PI(4,5)P$_2$ liposomes in the presence of both other proteins. [Nwk] = 1 μM, [WASp] = 1 μM, [Dap160^SH3CD] = 3 μM. (D) Giant unilamellar vesicle (GUV) decoration assay, with 10% PI(4,5)P$_2$ GUVs labeled with <1% TopFluor-PE. The addition of WASp to Nwk (red) and Dap160^SH3CD (blue) enhances the recruitment of Dap160^SH3CD to the membrane (green, note diffuse blue signal in (-) WASp condition). [Nwk] = 250 nM, [WASp] = 250 nM, [Dap160^SH3CD] = 1 μM. Scale bar is 10 μm. (E, F) Fluorescence recovery after photobleaching (FRAP) assay of endogenously labeled Nwk in control and C155-GAL4/UAS-Dicer-driven Dap160^RNAi neuromuscular junctions (NMJs). Images show individual medial optical sections of Airyscan confocal images at the indicated time point. Control Nwk signal shows membrane association (see strong peripheral signal) and slower recovery kinetics, while loss of Dap160 eliminates the strong peripheral accumulation of Nwk::GFP and increases the recovery kinetics of Nwk::GFP in the bleached region (dashed circles). Graph shows mean ± sem; n represents NMJs. Scale bar is 5 μm. Associated with *Figure 5—figure supplement 1*.

The online version of this article includes the following source data and figure supplement(s) for figure 5:

**Source data 1.** Source data for *Figure 5* and associated figure supplements.

**Figure supplement 1.** Membrane binding and bending by Dap160-WASp-Nwk.

rather than alter the inherent activity of Nwk (*Figure 5—figure supplement 1D*). These data indicate that Dap160-Nwk SH3-mediated interactions potentiate Nwk association with membranes in vitro.

Finally, to test whether Dap160 promotes Nwk membrane association in vivo, we examined the dynamics of Nwk at the synapse in the presence and absence of Dap160. Knockdown of Dap160 by RNAi (*Figure 5E*, *Figure 3—figure supplement 2D*) led to a striking loss of endogenously tagged Nwk::GFP from synaptic membranes (note strong peripheral labeling in control bouton cross-sections; *Figure 5E*). Further, Dap160 knockdown significantly increased the rate of recovery of Nwk::GFP after photobleaching, consistent with a shift in localization from membrane-bound to cytosolic (*Figure 5F*). These data suggest that the Dap160$^{SH3CD}$-Nwk interaction promotes Nwk membrane association in vivo. Taken together, our data indicate that multiple coordinated interactions between Nwk, WASp, Dap160$^{SH3CD}$, and membranes are required to relieve Nwk autoinhibition, allowing for tight control of membrane-coupled actin assembly in the PAZ.

## Dap160-Nwk interactions regulate synaptic F-actin patch dynamics

To determine how these mechanisms direct WASp-mediated actin assembly at the synapse, we measured actin dynamics in *nwk* (*Figure 6A–C*, *Video 2*) and *dap160* domain (*Figure 6D–F*) mutant NMJs. We predicted two possible but non-exclusive functions based on the dual roles that we found

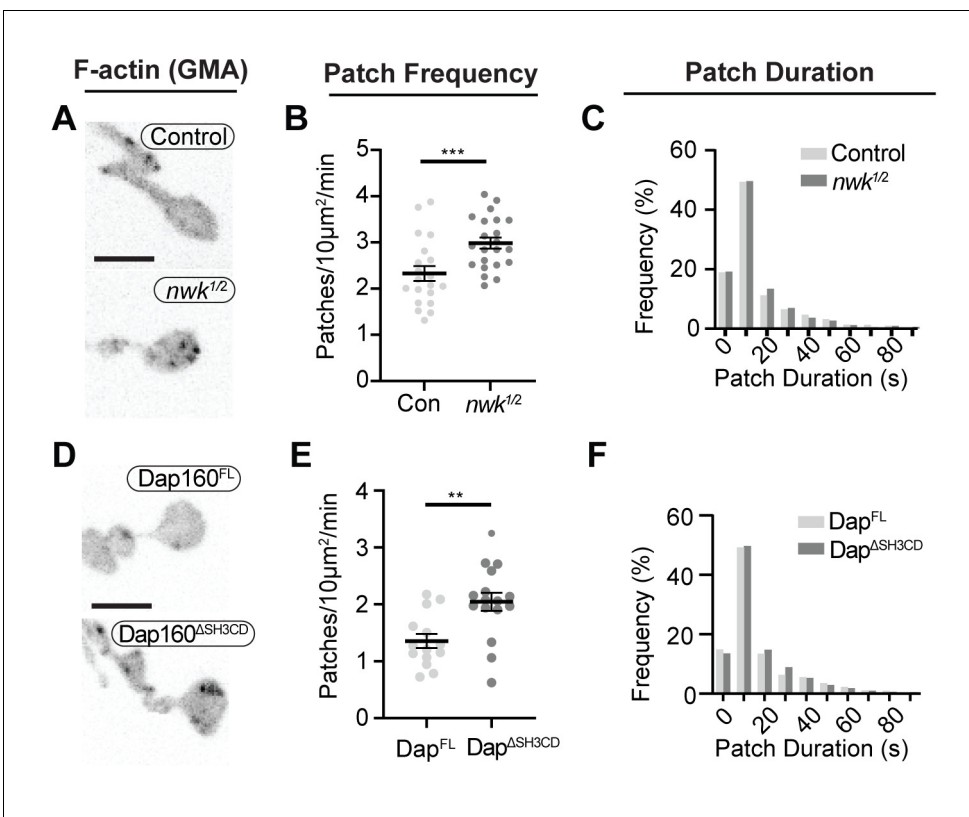

**Figure 6.** Loss of the Dap160-Nwk interaction disrupts actin patch dynamics at synapses in vivo. (**A, D**) Maximum intensity projections (MaxIPs) of live spinning disc confocal micrographs of presynaptically expressed GMA in muscle 6/7 neuromuscular junctions (NMJs) of the indicated genotypes, imaged at 1 Hz. Graphs quantify patch frequency (**B, E**) and distribution of patch durations (**C, F**). Loss of *nwk* (**A–C**) or of the Nwk-interacting Dap160$^{SH3CD}$ domain (**D–F**) increases the frequency of actin patch assembly. In both cases, there is no change in the distribution of patch durations (**C, F**). Scale bars in (**A, D**) are 5 µm. Associated with *Figure 6—figure supplements 1–2*, *Video 2*.

The online version of this article includes the following source data and figure supplement(s) for figure 6:

**Source data 1.** Source data for *Figure 6* and associated figure supplements.

**Figure supplement 1.** Analysis of actin dynamics is robust to tracking parameters at 1 Hz imaging.

**Figure supplement 2.** Validation of actin particle analysis in *nwk* mutants.

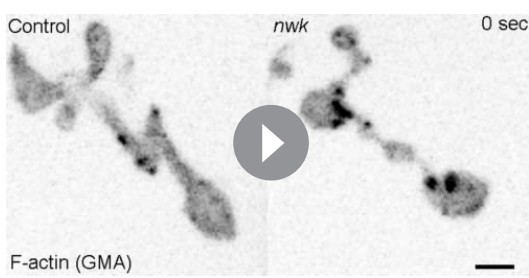

**Video 2.** Loss of *nwk* increases the frequency of brief actin patches.
https://elifesciences.org/articles/69597#video2

for the Nwk-Dap160-WASp module in vitro: if Nwk and Dap160 are primary activators of WASp, then loss-of-function mutants are likely to diminish patch frequency, duration, or intensity. Importantly, multiple WASp activators exist in the synaptic endocytic machinery (e.g., Cip4 and Snx9; *Almeida-Souza et al., 2018*; *Gallop et al., 2013*; *Nahm et al., 2010*; *Ukken et al., 2016*), and therefore, these could make significant contributions to WASp activation in addition to Nwk. Conversely, if an important function of autoinhibition is to 'clamp' actin assembly at the synapse, we expected that loss of Nwk and/or Dap160 would lead to spurious actin assembly events by these other WASp regulators. We found that both *nwk* and Dap160$^{\Delta SH3CD}$ mutants significantly increased patch frequency (*Figure 6B, E*, *Figure 6—figure supplement 1*), supporting a clamp function for these proteins. We did not detect a difference in the distribution of patch lifetimes, suggesting that it is the frequency of events, and not their duration per se, that changes (*Figure 6C,F*).

We also analyzed actin dynamics using a complementary approach in which we measured the normalized intensity variation (coefficient of variation, CoV) over time across the entire NMJ. Interestingly, the magnitude of variation was significantly higher in *nwk* mutants (*Figure 6—figure supplement 2A–B*), but the area of the NMJ that was highly variant was similar between genotypes, suggesting that actin assembly is more dynamic in time in these mutants, rather than more extensive in space (*Figure 6—figure supplement 2C*). We validated this analysis for its sensitivity in detecting changes in event frequency by analyzing synthetic data (*Figure 6—figure supplement 2D*, see 'Materials and methods' for details). The modeled data suggest that the difference in CoV that we measured between Control and *nwk* is consistent with a 43% increase in patch frequency, which is slightly higher than our measurement by particle tracking (28%; *Figure 6A*). This complementary analysis does not rely on particle tracking and makes no assumptions about the nature of actin dynamics, and is consistent with our particle-based metrics. Thus, we conclude that these phenotypes are robust to the method of analysis used.

## Nwk and Dap160$^{SH3CD}$ are required for normal synaptic vesicle endocytosis

We next investigated the physiological function of actin patches in vivo. Considering that patch morphology, frequency, and duration resembled endocytic dynamics, we first compared actin patches with the endocytic adaptors Clc and AP2α. Like other endocytic proteins, both presynaptically expressed Clc::GFP and endogenously tagged AP2α::GFP were primarily enriched at the plasma membrane relative to the cytoplasm (*Figure 7—figure supplement 1A*) and covered a large area fraction of the membrane, similar to other endocytic proteins (*Figure 2*). In addition to diffuse signal, both probes localized to short- and long-lived puncta, a subset of which dynamically colocalized with actin patches (*Figure 7A–C*, *Figure 7—figure supplement 1B*, *Video 3*). A significant proportion of endogenously labeled AP2 at the NMJ is likely associated with the closely apposed postsynaptic membrane, which accounts for its slightly lower correlation coefficient with Lifeact::Ruby. Considering that the rates of exo/endocytosis at this synapse at rest are relatively low (see above), these observations suggest that like other PAZ endocytic proteins, a large pool of membrane-localized clathrin coat and adaptor proteins are not actively engaged in endocytosis. Despite these caveats, we found that actin significantly colocalized with both Clc (*Figure 7C*) and AP2 (*Figure 7—figure supplement 1C*), consistent with a role in endocytosis for these actin-enriched sites. To more rigorously and functionally test the hypothesis that actin patches are endocytic, we acutely disrupted endocytic dynamics using the temperature-sensitive dominant-negative dynamin/*shi*$^{TS1}$ allele. When imaged under restrictive conditions, *shi* disruption decreased the frequency of actin patch dynamics (*Figure 7D–E*). Together, these data suggest that a significant fraction of presynaptic actin patches are associated with endocytosis.

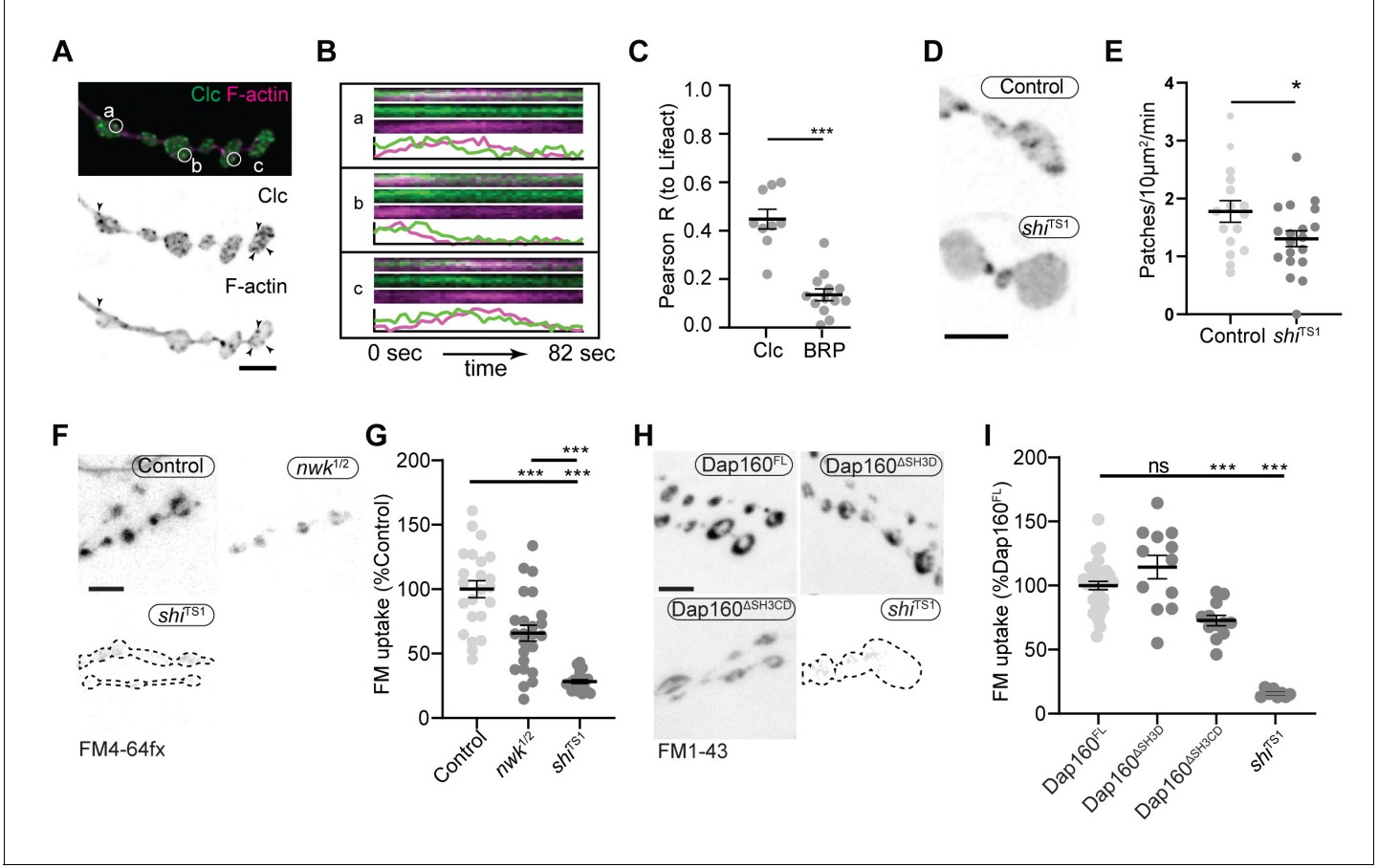

**Figure 7.** Actin patches and the Nwk-Dap160 interaction are associated with synaptic endocytosis. (A, B) Sum intensity projection (A) and representative kymographs (B) of spinning disc confocal timelapse of presynaptically expressed Lifeact::Ruby and clc::GFP. (A) Sum projection of 41 frames (82 s) highlights overlapping intensities of clc and Lifeact (circles and arrowheads). Circles indicate locations of kymographs in panel (B). (B) Kymographs of clc and Lifeact signals. Kymographs span the full duration of the movie from left (0 s) to right (82 s). Intensity profiles were normalized per channel from the minimum to the maximum value of each profile. (C) Quantification of colocalization between Lifeact::Ruby and Clc::GFP. Presynaptically expressed Lifeact::Ruby was co-expressed with either presynaptically expressed Clc::GFP or a BRP::GFP knockin and imaged in 3D stacks by Airyscan. Bruchpilot (BRP) control is the same dataset as in *Figure 1F* (all data were acquired contemporaneously). (D, E) Normal patch assembly requires dynamin activity. (D) Maximum intensity projections (MaxIPs) of single spinning disc confocal microscopy time points, showing pan-neuronally expressed GFP::actin in control and $shi^{TS1}$ mutant muscle 6/7 neuromuscular junctions (NMJs), imaged at 1 Hz, at the restrictive temperature of 31°C under stimulating conditions to drive the terminal $shi^{TS1}$ phenotype (45 mM KCl, 2 mM CaCl$_2$). Graph shows mean ± sem. n represents NMJs. (E) Quantification of patch frequency. (F–I) FM dye uptake assays at muscle 6/7 NMJs following 5 min of 90 mM potassium stimulation at 36°C. (F, H) MaxIPs of spinning disc confocal micrographs of FM dye uptake assays. Dotted lines highlight synapse outline in $shi^{TS1}$ NMJs. (F, G) *nwk* mutants exhibit partially defective FM4-64fx dye uptake relative to $shi^{TS1}$ mutants. (H, I) Loss of Dap160-Nwk interactions in a Dap160$^{\Delta SH3CD}$ truncation (but not Dap160$^{\Delta SH3D}$) exhibits partially defective FM1-43 dye uptake relative to $shi^{TS1}$, similar to *nwk* mutants. Graphs show mean ± sem; n represents NMJs. Scale bars are 2.5 µm (A, B) or 5 µm (D, F, H). Associated with *Figure 7—figure supplements 1–3*, *Video 3*.

The online version of this article includes the following source data and figure supplement(s) for figure 7:

**Source data 1.** Source data for *Figure 7* and associated figure supplements.

**Figure supplement 1.** Colocalization of actin with AP2α and Clathrin light chain.

**Figure supplement 2.** All Dap160 transgenes rescue dap160 satellite bouton phenotype.

**Figure supplement 3.** *nwk* and *dap160* domain mutants do not disrupt FM dye unloading.

We next tested the physiological requirement of the Nwk and Dap160$^{SH3CD}$ interaction. As both Nwk and Dap160 are implicated in the endocytic trafficking of synaptic growth-promoting bone morphogenetic protein (BMP) receptors (*O'Connor-Giles et al., 2008*; *Rodal et al., 2008*), we tested whether the Dap160-Nwk interaction was required for normal synaptic growth, which we assayed by counting satellite boutons, a hallmark phenotype of both null mutants. Surprisingly, we found that both Dap160$^{\Delta SH3D}$ and Dap160$^{\Delta SH3CD}$ truncations rescued satellite bouton numbers to

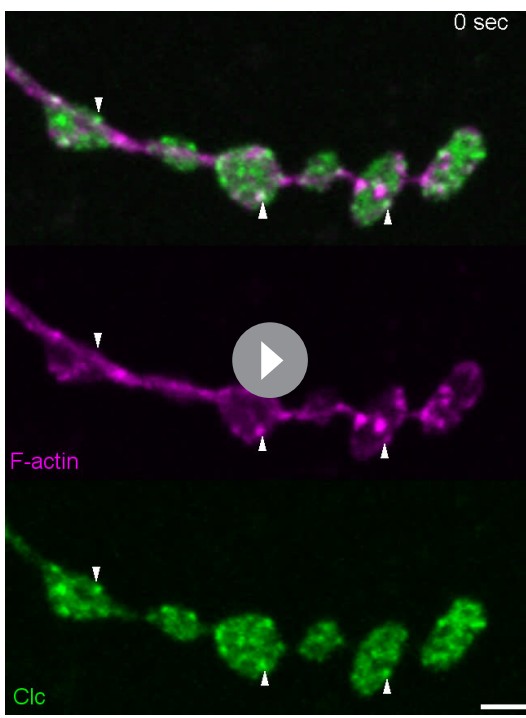

**Video 3.** Clc-GFP and Lifeact::Ruby partially colocalize.
https://elifesciences.org/articles/69597#video3

wild-type levels (*Figure 7—figure supplement 2*). These data indicate that synaptic vesicle and growth factor endocytosis are mechanistically separable, and suggest that actin dynamics phenotypes in the Dap160$^{\Delta SH3CD}$ mutant are not associated with synaptic growth regulation. We next examined synaptic vesicle endocytosis and recycling by FM dye uptake. *nwk*$^{1/2}$ null mutants led to a 34% decrease in FM4-64fx uptake compared to controls (*Figure 7F–G*), an intermediate phenotype compared to dominant negative dynamin in *shi*$^{TS1}$ mutants (72% decrease). *dap160* null mutants have been previously shown to exhibit an endocytosis defect (*Koh et al., 2004*; *Marie et al., 2004*), so we next tested whether the interaction between Dap160 and Nwk is required to support normal endocytosis. Indeed, we found that expression of Dap160$^{\Delta SH3CD}$ in *dap160* null mutants also significantly diminished FM dye uptake to a similar extent as loss of *nwk* (27% reduction; *Figure 7H–I*). By contrast, loss of the Dap160$^{SH3D}$ domain alone caused no defects in FM uptake, consistent with the lack of effect of this mutation on Nwk accumulation and localization (*Figure 3D–E*), suggesting that this interaction, though required in vitro, may be compensated by additional factors in vivo. Both *nwk* and Dap160$^{\Delta SH3CD}$ mutants unloaded FM dye to the same extent as controls, suggesting that diminished endocytosis is a direct phenotype, and not secondary to exocytic deficits (*Figure 7—figure supplement 3*). Importantly, these data indicate that spurious actin assembly events in *nwk* and *dap160* mutants are likely to be unproductive for normal endocytosis. Overall, our data support the hypothesis that normal synaptic actin patches represent active endocytic events and indicate that Dap160-Nwk regulation of actin patch dynamics is functionally required for synaptic vesicle endocytosis.

## Discussion

Here we have identified a mechanism by which autoinhibition clamps the presynaptic endocytic machinery to regulate the dynamics of discrete synaptic actin assembly events and the efficiency of synaptic endocytosis. Our data suggest a model in which specific interactions among Nwk, Dap160, and WASp function in two ways to potentiate membrane-associated actin dynamics. (1) Persistent autoinhibition of Nwk allows for stable binding of inactive PAZ machinery to presynaptic membranes to constrain spurious actin assembly events. (2) Coordinated relief of Nwk autoinhibition by Dap160 and WASp robustly activates F-actin assembly and ensures that actin assembles into structures that are likely to productively remodel membranes. This provides a mechanism by which synapses can use the micron-scale PAZ organization of endocytic machinery as a regulated reservoir to efficiently generate 50- to 100-nm-scale endocytic events, in response to physiological cues such as synaptic transmission.

### Predominant presynaptic actin structures resemble endocytic patches

Here we provide the first quantitative analysis of the composition and dynamics of individual presynaptic F-actin structures. Numerous studies have examined actin dynamics at the level of entire synapses or qualitatively described dynamics of discrete actin structures (*Bloom et al., 2003*; *Colicos et al., 2001*; *Nunes et al., 2006*; *Piccioli and Littleton, 2014*; *Sankaranarayanan et al., 2003*; *Zhao et al., 2013*), and identified diverse roles for actin, including synaptic vesicle endocytosis (*Holt et al., 2003*; *Kononenko et al., 2014*; *Soykan et al., 2017*; *Watanabe et al., 2013*; *Wu et al.,*

2016; *Zhao et al., 2013*), synaptic vesicle organization and mobilization (*Guzman et al., 2019*; *Lee et al., 2012*; *Marra et al., 2012*; *Owe et al., 2009*; *Sakaba and Neher, 2003*; *Wolf et al., 2015*), active zone organization and function (*Pilo Boyl et al., 2007*; *Morales et al., 2000*; *Wagh et al., 2015*; *Waites et al., 2011*; *Wang et al., 1999*), and receptor-mediated endocytosis (*Kim et al., 2019*; *Rodal et al., 2008*). Bulk analyses, which do not separate individual dynamic actin structures in space and time, are limited in their ability to discern how the regulation and dynamics of actin contribute to these distinct functions. We leveraged our ability to extract data describing individual structures to find that synaptic actin predominantly assembled into discrete Arp2/3-associated patches, and identified points of control over their dynamics. Specifically, we found that loss of endocytic proteins differentially affected the frequency and kinetics of individual actin patches, which correlate with functional deficits in endocytosis.

The link between the actin structures that we observed and endocytic events is supported by several lines of evidence: the morphology and duration of synaptic actin patches are similar to WASp/Arp2/3-dependent endocytic actin dynamics in yeast (16 s; *Berro and Pollard, 2014*) and somewhat briefer than in cultured mammalian cells (~40 s; *Grassart et al., 2014*; *Merrifield et al., 2004*; *Taylor et al., 2011*). Given the measured time constant for *endocytosis* (~14 s; *Poskanzer et al., 2006*) and clathrin dependence of vesicle cycling in this synapse (*Heerssen et al., 2008*), these values support the hypothesis that synaptic actin patches are likely sites of clathrin-mediated endocytosis. Further, the frequency of patch assembly, which we measured in resting synapses, approaches the rate of spontaneous synaptic vesicle release at this synapse (*Figure 1—figure supplement 1B*, *Figure 1—figure supplement 2A*) (~5–6/10 $\mu m^2$/min; *Melom et al., 2013*; *Akbergenova et al., 2018*). Further, actin patches colocalize partially with endocytic adaptors, and their assembly is sensitive to disruption of endocytosis (*Figure 7*). Finally, we found that the same endocytic proteins and protein interactions that regulate endocytosis at this synapse also alter the dynamics of actin patches.

Technical challenges due to the high density of endocytic proteins and synaptic vesicle cargoes, and the difficulty of conducting sparse single vesicle measurements at this synapse (compared to neurons in culture; *Chanaday and Kavalali, 2018*; *Peng et al., 2012*), make it difficult to directly link the dynamics of actin structures to specific membrane or cargo internalization events. However, the frequency of the events captured by our approach makes it unlikely that they represent rare F-actin-dependent events at this synapse, such as those that control macropinocytosis or new bouton growth (*Khuong et al., 2010*; *Kim et al., 2019*; *Piccioli and Littleton, 2014*), and more likely that they represent bona fide endocytic events. Thus, while we do not rule out other biological functions for a subset of patches, together our data indicate that a significant and measurable fraction of synaptic actin patches are associated with endocytosis.

## Autoinhibition clamps PAZ membrane-remodeling machinery at synapses

Many endocytic proteins accumulate across broad membrane domains at the *Drosophila* NMJ and other synapses (*Gerth et al., 2017*; *Guichet et al., 2002*; *Koh et al., 2007*; *Roos and Kelly, 1998*; *Verstreken et al., 2008*, *Verstreken et al., 2003*). Our data indicate that much of this membrane-remodeling machinery is likely held in an inactive state at the presynaptic membrane: Nwk and Dap160 accumulate in a micron-scale membrane domain (*Figure 2*), and their loss increases the frequency of short-lived actin patches (*Figure 6*). These data suggest that these PAZ proteins are held in a partially autoinhibited state at the membrane in vivo, consistent with our prior in vitro observations (*Stanishneva-Konovalova et al., 2016*). This is further consistent with the broad distribution of Clc and AP2 to the membrane (*Figure 7*, *Figure 7—figure supplement 1*). Given the comparatively low rate of endocytosis expected at rest at this synapse, this suggests that most Clc and AP2 puncta at the synapse are either not stabilized to form productive endocytic sites (*Aguet et al., 2013*) or associated with some non-endocytic functions (*Gimber et al., 2015*).

The fact that loss of Nwk increases the frequency of patches while decreasing FM uptake suggests that the actin structures assembled in the *nwk* mutant are unproductive for synaptic vesicle endocytosis. These spurious patches could reflect non-specific actin assembly, perhaps due to unmasking of the Nwk ligand PI(4,5)$P_2$ at the membrane and/or inappropriate activation of alternative WASp-dependent actin assembly pathways. Indeed, additional WASp activators such as Snx9 and Cip4/Toca-1 may play accessory roles in endocytic actin assembly (*Almeida-Souza et al., 2018*;

*Gallop et al., 2013*), consistent with our finding that loss of presynaptic WASp leads to a decrease in the total number of patches (*Figure 1G–I*, *Figure 1—figure supplement 1C–E*). Our data indicate that at the synapse, where endocytic machinery accumulates at high concentrations (*Wilhelm et al., 2014*) and recruitment appears uncoupled from activation, these layers of autoregulation are critical to constrain actin assembly generally.

Our findings on autoinhibition and clamping connect two prevailing models of the organization and function of the synaptic endocytosis machinery—preassembly and delivery. In the first model, preassembly of clathrin and accessory proteins is hypothesized to ensure fast endocytosis in response to synaptic vesicle fusion (*Hua et al., 2011*; *Mueller et al., 2004*; *Wienisch and Klingauf, 2006*). Here, Nwk autoinhibition provides a mechanism to assemble an inactive, yet poised endocytic apparatus. In the second model, endocytic machinery associates with the synaptic vesicle pool, providing a ready source or buffer of proteins that can be released to the plasma membrane upon calcium signaling or vesicle fusion (*Bai et al., 2010*; *Denker et al., 2011*; *Gerth et al., 2017*; *Winther et al., 2015*). Because Dap160/intersectin can shuttle between the synaptic vesicle pool and the plasma membrane, is itself subject to autoregulation (*Gerth et al., 2017*), and can regulate other endocytic proteins (e.g., dynamin, Nwk), it could serve as a single activator that couples the preassembly and delivery models.

## Coordinated relief of autoinhibition directs membrane-associated actin dynamics

Our in vitro data show that beyond functioning as a clamp, Nwk and Dap160 collaboratively activate WASp to promote robust actin assembly. Together with the defects we observed in vivo for actin dynamics and FM dye uptake, these data suggest that Dap160-Nwk-WASp interactions could serve as a coincidence detection mechanism to relieve autoinhibition of Nwk and promote productive actin assembly with other WASp activators. Coincidence detection has been demonstrated in several systems to control membrane-associated actin assembly (*Case et al., 2019*; *Sun et al., 2017*), suggesting that amplification of WASp membrane binding could drive robust actin patch assembly at synapses. Similarly, in human cells, the interaction between FCHSD2, intersectin, and WASp promotes actin assembly and endocytic maturation (*Almeida-Souza et al., 2018*) or initiation (*Xiao et al., 2018*). The Dap160-Nwk module could act by directing and/or organizing actin assembly specifically at endocytic events, akin to the membrane-directed actin assembly we observed in vitro (*Figure 5D*), and/or ensure that it is sufficiently robust for productive membrane remodeling (*Akamatsu et al., 2020*). Direct support for these models will require new analytical or imaging approaches to directly visualize the coupling of membranes and actin to the endocytic machinery, in order to distinguish spurious (due to unclamping) vs bona fide but underpowered endocytic actin assembly events.

## Physiological implications of autoregulatory mechanisms in the periactive zone

Our data suggest that the endocytic machinery can be deployed as clamped, primed, or activated complexes at different locations at the synapse. The next critical step will be to determine the mechanisms that control switching between these states. Many potential mechanisms that link calcium-dependent exocytosis and endocytosis could activate actin assembly. These include direct effects of calcium on the endocytic machinery (*Maritzen and Haucke, 2018*), the accumulation of synaptic vesicle cargoes (*Cousin, 2017*), stoichiometry-dependent changes in protein interactions or activities (*Case et al., 2019*), changes in membrane mechanics (*Anantharam et al., 2010*; *Dai et al., 1997*; *Roux et al., 2010*), and changes in membrane charge/mode of membrane binding (*Kelley et al., 2015*). One intriguing possibility is that these mechanisms might enable an endocytic PAZ to rapidly switch between different modes of endocytosis (e.g., ultrafast, conventional, or bulk) in response to a wide range of synaptic activity patterns (*Gan and Watanabe, 2018*). These endocytic regulatory mechanisms could also be locally poised to regulate, respond, or adapt to the specific release properties of nearby active zones (*Akbergenova et al., 2018*; *Melom et al., 2013*; *Dickman et al., 2006*), and serve as novel points of control over synaptic plasticity and homeostasis.

# Materials and methods

## Key resources table

| Reagent type (species) or resource | Designation | Source or reference | Identifiers | Additional information |
|---|---|---|---|---|
| Gene (*Drosophila melanogaster*) | *nwk* | GenBank | FLYB: FBgn0263456 | |
| Gene (*D. melanogaster*) | *dap160* | GenBank | FLYB: FBgn0023388 | |
| Gene (*D. melanogaster*) | *wsp* | GenBank | FLYB: FBgn0024273 | |
| Gene (*D. melanogaster*) | *shi* | GenBank | FLYB: FBgn0003392 | |
| Gene (*D. melanogaster*) | *clc* | GenBank | FLYB: FBgn0024814 | |
| Gene (*D. melanogaster*) | *AP-2α* | GenBank | FLYB: FBgn0264855 | |
| Genetic reagent (*D. melanogaster*) | AP2α::GFP | This study | | Maintained in Kaksonen Lab - see 'Methods' for description |
| Genetic reagent (*D. melanogaster*) | w; UAS-WASp::TEV-Myc IIB (chromosome II insertion) | This study | | Maintained in Rodal Lab - see 'Methods' for description |
| Genetic reagent (*D. melanogaster*) | w; UAS-GFP::Moesin Actin Binding Domain (GMA) | Bloomington *Drosophila* Stock Center | BDSC:31776; FLYB: FBti0131132; RRID:BDSC_31777 | FlyBase symbol: P{UAS-GMA}3 |
| Genetic reagent (*D. melanogaster*) | w; UAS-Lifeact::Ruby | Bloomington *Drosophila* Stock Center | BDSC:35545; FLYB: FBst0035545; RRID:BDSC_35545 | FlyBase symbol: P{UAS-Lifeact-Ruby}VIE-19A |
| Genetic reagent (*D. melanogaster*) | w; UAS-Arp3::GFP | Bloomington *Drosophila* Stock Center | BDSC: 39722; FLYB: FBst0039722; RRID:BDSC_39722 | FlyBase symbol: P{w[+mC]=UASp-Arp3.GFP}3 |
| Genetic reagent (*D. melanogaster*) | w; $e^1$, $wsp^1$/TM6b,Tb | Bloomington *Drosophila* Stock Center | BDSC: 51657; FLYB: FBst0051657; RRID:BDSC_51657 | FlyBase symbol: e[1] WASp[1] |
| Genetic reagent (*D. melanogaster*) | w; UAS-GFP::actin | Bloomington *Drosophila* Stock Center | BDSC: 9258; FLYB: FBst0009258; RRID:BDSC_9258 | FlyBase symbol: P{w[+mC]=UASp-GFP.Act5C}2-1 |
| Genetic reagent (*D. melanogaster*) | yv; P{TRiP.HMC03360} attP40 - Wasp RNAi | Bloomington *Drosophila* Stock Center | BDSC: 51802; FLYB: FBst0051802; RRID:BDSC_51802 | FlyBase symbol: P{y[+t7.7] v[+t1.8]=TRiP.HMC03360}attP40 |
| Genetic reagent (*D. melanogaster*) | yw; UAS-luciferase RNAi | Bloomington *Drosophila* Stock Center | BDSC: 31603; FLYB: FBst0031603; RRID:BDSC_31603 | FlyBase symbol: P{y[+t7.7] v[+t1.8]=TRiP.JF01355}attP2 |
| Genetic reagent (*D. melanogaster*) | w; UAS-Dap 160$^{\Delta SH3D}$::mCherry VK00027 | This study | | Maintained in Rodal Lab - see 'Methods' for description |
| Genetic reagent (*D. melanogaster*) | w; UAS-Dap 160$^{\Delta SH3CD}$::mCherry VK00027 | This study | | Maintained in Rodal Lab - see 'Methods' for description |
| Genetic reagent (*D. melanogaster*) | w; UAS-Dap160$^{FL}$::mCherry VK00027 | This study | | Maintained in Rodal Lab - see 'Methods' for description |

*Continued on next page*

*Continued*

| Reagent type (species) or resource | Designation | Source or reference | Identifiers | Additional information |
|---|---|---|---|---|
| Genetic reagent (*D. melanogaster*) | yw; UAS-Dap160-RNAi | Bloomington Drosophila Stock Center | BDSC: 25879; FLYB: FBst0025879; RRID:BDSC_25879 | FlyBase symbol: P{y[+t7.7] v[+t1.8]=TRiP.JF01918}attP2 |
| Genetic reagent (*D. melanogaster*) | yw; Mi{PT-GFSTF.1} nwkMI05435-GFSTF.1 | Bloomington Drosophila Stock Center | BDSC: 64445; FLYB: FBst0064445; RRID:BDSC_64445 | FlyBase symbol: Mi{PT-GFSTF.1}nwk [MI05435-GFSTF.1] |
| Genetic reagent (*D. melanogaster*) | w; *nwk²,h* | *Coyle et al., 2004* | FLYB: FBal0154818 | FlyBase symbol: nwk[2] |
| Genetic reagent (*D. melanogaster*) | w; *nwk¹* | Bloomington Drosophila Stock Center | BDSC: 51626; FLYB: FBst0051626; RRID:BDSC_51626 | FlyBase symbol: nwk[1] |
| Genetic reagent (*D. melanogaster*) | w; *dap160^{Δ1}* | Bloomington Drosophila Stock Center | BDSC: 24877; FLYB: FBst0024877; RRID:BDSC_24877 | FlyBase symbol: Dap160[Delta1] |
| Genetic reagent (*D. melanogaster*) | w; Df(2L)Exel6047, P{XP-U}Exel6047/ CyOGFP (Dap160^{Df}) | Bloomington Drosophila Stock Center | BDSC: 7529; FLYB: FBst0007529; RRID:BDSC_7529 | FlyBase symbol: Df(2L)Exel6047, P{w[+mC]=XP-U}Exel6047 |
| Genetic reagent (*D. melanogaster*) | *dvglut*(X)-GAL4 | *Daniels et al., 2008* | FLYB: FBti0129146 | FlyBase symbol: P{VGlut-GAL4.D}NMJX |
| Genetic reagent (*D. melanogaster*) | *elav^{c155}*-GAL4 | Bloomington Drosophila Stock Center | BDSC: 458; FLYB: FBst0000458; RRID:BDSC_458 | FlyBase symbol: P{w[+mW.hs]= GawB}elav[C155] |
| Genetic reagent (*D. melanogaster*) | UAS-Dcr2 | Bloomington Drosophila Stock Center | BDSC: 24646; FLYB: FBst0024646; RRID:BDSC_24646 | FlyBase symbol: P{w[+mC]= UAS-Dcr-2.D}1 |
| Genetic reagent (*D. melanogaster*) | CD8-mCherry | Bloomington Drosophila Stock Center | BDSC:32218; FLYB: FBst0032218; RRID:BDSC_32218 | FlyBase symbol: P{y[+t7.7] w[+mC]= 10XUAS-IVS-mCD8::RFP}attP2 |
| Antibody | Rabbit α-Nwk Polyclonal | Coyle 2004 | #970 RRID:AB_2567353 | IF(1:1000), WB (1:1000) |
| Antibody | Mouse α-Brp Monoclonal | DSHB | RRID:AB_2314866 | IF(1:100) |
| Antibody | Mouse α-Myc Monoclonal | DSHB | RRID:AB_2266850 | IF(1:50) |
| Antibody | Rabbit α-Dap160 Polyclonal | Davis/Kelly | RRID:AB_2569367 | IF(1:1000) |
| Antibody | Mouse α-Xpress Monoclonal | ThermoFisher | RRID:AB_2556552 | WB(1:1000) |
| Antibody | Mouse α-Tubulin Monoclonal | Sigma | RRID:AB_477579 | WB(1:1000) |
| Antibody | α-HRP Polyclonal | Jackson Immuno Research | RRID:AB_2338967 | IF(1:500) |
| Recombinant DNA reagent | His-Nwk^{607-731} | *Kelley et al., 2015* | | |
| Recombinant DNA reagent | GST | *Kelley et al., 2015* | | |
| Recombinant DNA reagent | 6His-Dap160^{SH3C} | This study | | Maintained in Rodal Lab - see 'Methods' for description |
| Recombinant DNA reagent | 6His-Dap160^{SH3D} | This study | | Maintained in Rodal Lab - see 'Methods' for description |

*Continued on next page*

*Continued*

| Reagent type (species) or resource | Designation | Source or reference | Identifiers | Additional information |
|---|---|---|---|---|
| Recombinant DNA reagent | 6His-Dap160$^{SH3CD}$ | This study | | Maintained in Rodal Lab - see 'Methods' for description |
| Recombinant DNA reagent | GST-Dap160$^{SH3C}$ | This study | | Maintained in Rodal Lab - see 'Methods' for description |
| Recombinant DNA reagent | GST-Dap160$^{SH3D}$ | This study | | Maintained in Rodal Lab - see 'Methods' for description |
| Recombinant DNA reagent | GST-Dap160$^{SH3CD}$ | This study | | Maintained in Rodal Lab - see 'Methods' for description |
| Recombinant DNA reagent | His-Nwk$^{1-428}$ | *Becalska et al., 2013* | | |
| Recombinant DNA reagent | His-Nwk$^{1-633}$ | *Kelley et al., 2015* | | |
| Recombinant DNA reagent | His-Nwk$^{1-731}$ | *Kelley et al., 2015* | | |
| Recombinant DNA reagent | His-WASp$^{1-143}$ | *Rodal et al., 2008* | | |
| Recombinant DNA reagent | His-SNAP-Nwk$^{1-731}$ | *Kelley et al., 2015* | | |
| Recombinant DNA reagent | His-SNAP-Nwk$^{1-633}$ | This study | | Maintained in Rodal Lab - see 'Methods' for description |
| Recombinant DNA reagent | His-SNAP-Dap160$^{SH3CD}$ | This study | | Maintained in Rodal Lab - see 'Methods' for description |
| Sequence-based reagent | UAS-Dap160SH3ΔCD-FWD | This paper | PCR primers | ATGAACTCGGCGG TGGATGCGTGG |
| Sequence-based reagent | UAS-Dap160SH3ΔCD-REV | This paper | PCR primers | CCACATCAGCCTT TTGGACAT |
| Sequence-based reagent | UAS-Dap160SH3ΔD-FWD | This paper | PCR primers | ATGAACTCGGCGG TGGATGCGTGG |
| Sequence-based reagent | UAS-Dap160SH3ΔD-REV | This paper | PCR primers | GAGAACCTTCAC GTAAGTGGC |
| Sequence-based reagent | UAS-Dap160SH3FL-FWD | This paper | PCR primers | ATGAACTCGGCGG TGGATGCGTGG |
| Sequence-based reagent | UAS-Dap160SH3FL-REV | This paper | PCR primers | TCTTCTTGGTG GTGCCATTTG |
| Sequence-based reagent | His/GST-Dap160$^{SH3C}$-FWD | This paper | PCR primers | GGAATGCGTGC CAAGCGG |
| Sequence-based reagent | His/GST-Dap160$^{SH3C}$-REV | This paper | PCR primers | TTGGAGAACCT TCACGTAAGTGG |
| Sequence-based reagent | His/GST-Dap160$^{SH3CD}$-FWD | This paper | PCR primers | GGAATGCGTGC CAAGCGG |
| Sequence-based reagent | His/GST-Dap160$^{SH3CD}$-REV | This paper | PCR primers | TCACTTCTTGG TGGTGCCATTTGC |
| Sequence-based reagent | His/GST-Dap160$^{SH3D}$-FWD | This paper | PCR primers | CAAGGTCATTG CTCTCTATCCG |
| Sequence-based reagent | His/GST-Dap160$^{SH3D}$-REV | This paper | PCR primers | TCACTTCTTGGT GGTGCCATTTGC |

*Continued on next page*

*Continued*

| Reagent type (species) or resource | Designation | Source or reference | Identifiers | Additional information |
|---|---|---|---|---|
| Sequence-based reagent | His/GST-Dap 160$^{SH3ABCD}$-FWD | This paper | PCR primers | CACAGGCTCTTC CAGTGCTTGG |
| Sequence-based reagent | His/GST-Dap 160$^{SH3ABCD}$-REV | This paper | PCR primers | TCACTTCTTGGTG GTGCCATTTGC |
| Peptide, recombinant protein | Arp2/3 Complex | Cytoskeleton, Inc | RP01-P | |
| Biological sample (*Oryctolagus cuniculus*) | Rabbit Muscle | Pel-Freez | 41225 -2 | |
| Software, algorithm | Prism | Graphpad | RRID:SCR_002798 | |
| Software, algorithm | FIJI | FIJI | RRID:SCR_002285 | |
| Other | DOPC | Echelon | 1182 | |
| Other | DOPS | Avanti | 840035C | |
| Other | PI(4,5)P$_2$ | Avanti | 840046X | |
| Other | TopFluor-PE | Avanti | 810282C | |
| Other | DOPE | Echelon | 2182 | |
| Other | FM1-43FX | ThermoFisher | F35355 | |
| Other | FM4-64FX | ThermoFisher | F34653 | |

## Resource availability

### Lead contact

Further information and requests for resources and reagents should be directed to and will be fulfilled by the Lead Contact, Avital Rodal (arodal@brandeis.edu).

### Material availability

All plasmids and fly lines generated in these studies are available upon request.

## Experimental model

### *Drosophila* culture

Flies were cultured using standard media and techniques. All flies were raised at 25°C, with the exception of experiments using Dcr2; Dap160 RNAi or WASp RNAi, for which flies were raised at 29°C. See 'Key resources' table for all fly lines used and see *Supplementary file 1* for full genotypes for each experiment in this study.

## Methods

### Cloning

UAS-Dap160 constructs were generated in pBI-UASC-mCherry (derived from *Wang et al., 2011*; see *Deshpande et al., 2016*). Fragments were amplified from the genomic Dap160 locus with primers described in the 'Key resources' table. These transgenes were injected into flies (Rainbow Gene), using ΦC31-mediated integration at the VK00027 locus (*Venken et al., 2006*), to ensure that all constructs were in a similar genomic context. UAS-WASp-tev-myc was generated in pUAST (*Brand and Perrimon, 1993*) by inserting a Tobacco Etch Virus protease recognition site and nine copies of the myc epitope tag at the 3' end of the Wsp cDNA, and injected into w[1118] flies at the Duke Model Systems Transgenic Facility (Duke University, Durham, NC).

## Generation of AP2α::GFP[KI]

The vector pHD-sfGFP-dsRed was created using Gibson assembly by amplifying sfGFP from pScarlessHD-sfGFP-DsRed (gift from Kate O'Connor-Giles, Addgene plasmid # 80811) and inserting it in between the multiple cloning site and the first loxP site in the pHD-DsRed backbone (gift from Kate O'Connor-Giles, Addgene plasmid # 51434). 1 kb sequences upstream and downstream of the stop codon were amplified from the genomic locus of AP2α and inserted into pHD-sfGFP-dsRed using AarI and SapI, respectively, to create the HDR donor pMM007_pHD-AP2a-C-sfGFP-dsRed. The guide RNA GGAAATCTGCGATCTGTTGA was cloned into pU6-BbsI-chiRNA (gift from Melissa Harrison, Kate O'Connor-Giles, and Jill Wildonger, Addgene plasmid # 45946; *Gratz et al., 2013*) using BbsI to create pMM008_pU6-AP2a-chiRNA. 500 ng/ul HDR donor plasmid and 100 ng/ul gRNA plasmid were injected into vas-Cas9(III) flies (BDSC 51324, injections by BestGene). Correct integration of the transgene was validated by sequencing.

## FM dye uptake

FM dye (FM1-43 in *dap160* experiments or FM4-64FX in *nwk* experiments) uptake experiments were performed essentially as described (*Ramachandran and Budnik, 2010*; *Verstreken et al., 2008*). For fixed experiments (*nwk* mutants), larvae were dissected in groups of four to six (with each dish having at least two control larvae) in low-calcium HL3 (*Stewart et al., 1994*), and axons were cut to dissociate central nervous system input. For live imaging (*dap160* rescues), larvae were dissected, stained, and imaged in pairs, with one control (Dap160[FL]) and one experimental larva per dish. This temperature has been shown to exacerbate endocytic defects in some mutants, including *dap160* (*Koh et al., 2004*). Following extensive washing in Ca$^{++}$-free saline, larvae were fixed in 4% paraformaldehyde in Ca$^{++}$-free saline (for *nwk* experiments) or imaged live (for *dap160* transgene rescue experiments). Images of muscle 6/7 NMJs (abdominal segments 3–5) were acquired by confocal microscopy and FM dye intensity was measured within mCherry (in *dap160* experiments) or GFP (in *nwk* experiments)-labeled presynaptic masks, and intensities were normalized to dish-matched control larvae. For unloading experiments, larvae were analyzed individually. In all experiments, dye loading (4 μM) was performed in 90 mM KCl, 2 mM CaCl$_2$ HL3 saline for 5 min at 36°C on a submerged metal block using prewarmed buffer. For unloading, larvae were stimulated for an additional 5 min with 90 mM KCl, 2 mM CaCl$_2$, washed extensively in Ca$^{++}$-free HL3, then imaged and analyzed as for fixed larvae.

## Immunohistochemistry

For analysis of NMJ morphology and protein localization, flies were cultured at a low density at 25° C. Wandering third-instar larvae were dissected in calcium-free HL3.1 saline (*Feng et al., 2004*) and fixed for 30 min in HL3.1 containing 4% formaldehyde. For analysis of NMJ overgrowth (satellite boutons), samples were stained with α-HRP and α-Dlg (4F3) antibodies, and images were blinded before manual bouton counting. Boutons were counted on muscle 4 NMJs, abdominal segments 2–4, and satellite boutons were defined as any string of fewer than five boutons that branched from the main NMJ branch (*O'Connor-Giles et al., 2008*).

## Western blots

*Drosophila* heads (10 pooled heads/genotype) were homogenized in 100 μl 2x Laemmli buffer. 10 ul of extract per lane was fractionated by sodium dodecyl sulphate–polyacrylamide gel electrophoresis (SDS-PAGE) and immunoblotted with α-Dap160 (*Roos and Kelly, 1998*) and α-tubulin antibodies (clone B-5-1-2; Sigma), and infrared-conjugated secondary antibodies (Rockland, Inc). Blots were analyzed on a Biorad Chemidoc system.

## Imaging and analysis

Spinning disc confocal imaging of *Drosophila* larvae was performed at room temperature (except *shi*[TS1] experiments) on a Nikon Ni-E upright microscope equipped with 60x (NA 1.4) and 100x (NA 1.45) oil-immersion objectives, a Yokogawa CSU-W1 spinning disc head, and an Andor iXon 897U EMCCD camera. Images were collected using Nikon Elements AR software. For *shi*[TS1] GFP::actin imaging experiments, temperature was controlled using a CherryTemp temperature control unit (Cherry Biotech).

FRAP data (*Figure 5E*) were acquired on a Zeiss 880 microscope with Airyscan in super resolution acquisition mode, using a x63 NA 1.4 objective. Single Z-slices through the middle of individual boutons were acquired at 4 Hz for 90 s, with manual focus adjustment. Following acquisition of two or three initial Z-stacks to assess prebleach intensity, <20% of individual boutons were photobleached by the 488 laser at60% intensity and a scan speed of 6. Intensities of background, unbleached, and bleached ROIs were acquired manually using FIJI, and bleached area was normalized to prebleach and unbleached ROIs (to correct for imaging-induced photobleaching), and analyzed with GraphPad Prism.

Confocal imaging of GUVs and cell-sized water droplets was conducted at room temperature on a Marianas spinning disc confocal system (3I, Inc, Denver, CO), consisting of a Zeiss Observer Z1 microscope equipped with a Yokagawa CSU-X1 spinning disc confocal head, a QuantEM 512SC EMCCD camera, PLAN APOCHROMAT 63x or 100x oil-immersion objectives (NA 1.4), a Photonics Instruments Micropoint photo-ablation device, and Slidebook software.

3D-SIM was performed on a Nikon N-SIM E system (consisting of an inverted Eclipse Ti-E microscope, 100x (NA 1.45) oil-immersion objective, and a Hamamatsu OrcaFLASH4 sCMOS camera). Channel alignment was calculated for each imaging session using tetraspeck beads (Invitrogen, cat no. T-7284). Images were collected at room temperature with a regime of three grid orientations and five phases and were reconstructed using Nikon Elements software, using a theoretical, ideal optical transfer function generated by the software. Super-resolution images of protein localization in live samples were acquired with a Zeiss 880FAS microscope in fast Airyscan mode with a 63x (NA 1.4) oil-immersion objective, using Zen Blue software.

## Analysis of actin dynamics at the NMJ

Spinning disc confocal time series were acquired at 15 stacks/min (*Figure 1*), 60 stacks/min (*Figure 6*, *Figure 7E*), or 2.2 stacks/min (*Figure 7A*). A maximum intensity projection was made of each time point, videos were registered using the FIJI plugin StackReg, and analyzed by Patchtracker, based on Trackmate (*Berro and Pollard, 2014*) as follows. First, we qualitatively evaluated the optimal intensity threshold for patch detection by identifying the maximum threshold intensity at which all obvious patch structures were detected in the first frame of videos. This process was performed independently by three independent observers over multiple datasets. The threshold for patch detection was normalized to the mean probe intensity in the presynaptic area (threshold = Probe Mean * 0.32). All other settings for patch detection and tracking were default: estimated patch diameter = 0.6 µm, median filter = false, subpixel detection = true, linking max distance = 0.5 µm, gap-closing distance = 0.5 µm, gap-closing frame gap = 0. For 0.25 Hz imaging experiments, patches between 16 and 356 s could be detected. For 1 Hz imaging experiments, patches between 4 and 139 s could be detected. Because this analysis rejects a significant number of detected patches due to tracking defects or tracking path overlap, we estimated the true patch frequency as follows. We combined detections from 0.25 Hz and 1 Hz imaging experiments by averaging the frequencies over the shared detection range (20–150 s) and adding the lower and higher duration patches that were specific to each imaging regime (4–16 s for 1 Hz and 150–360 s for 0.25 Hz). Then we 'corrected' for rejected tracks and considered the lower bound of the estimate to be the actual, corrected merged frequency of detection (2.8 patches/10 µm/min) and the upper bound to include every rejected track (10.3 patches/10 µm/min).

We further validated our patch dynamics analysis by measuring patch frequencies at a wide range of patch intensity thresholds and track linking distances. For both 0.25 Hz WASp (*Figure 1—figure supplement 2*) and 1 Hz Nwk (*Figure 6—figure supplement 2*) datasets, we found measurements of control patch frequencies to be robust to these parameters and in strong agreement with the estimates described above across the entire parameter space tested (1.1–8.4 for 0.25 Hz imaging, 1.2–7.9 for 1 Hz imaging). Further, our phenotypic analyses (decreased patch frequency in WASp mutants and increased frequency in Nwk mutants) were also both highly robust to tracking parameters.

Actin dynamics were also analyzed by measuring intensity variation over time over the entire NMJ, that is, without thresholding or particle tracking. We measured this by extracting the intensity value for each pixel over time and calculating the CoV (Std Dev/Mean) for each pixel. We estimated the percentage of 'highly variant' pixels by thresholding these values using Li (*Li and Tam, 1998*) and Moments (*Tsai, 1995*) algorithms. While these two algorithms gave different estimates of the

fraction of NMJs covered by highly variant pixels, both indicated the same relationship between genotypes. To validate this approach, we created synthetic data using a custom FIJI script, with a spatial and temporal scale that matched our in vivo imaging, and in which we varied parameters expected to impact this metric (signal intensity, noise level, fraction of dynamic pixels, dynamics frequency, dynamics duration, dynamics amplitude), and subjected the synthetic data to the same CoV over time analysis.

## Intensity and colocalization analysis

For intensity and colocalization, the presynaptic region was masked in 3D using a presynaptically enriched label: either HRP (*Figure 3—figure supplement 2D*), Nwk (*Figure 2A*, *Figure 2C*), Dap160 (*Figure 2E*, *Figure 3D–E*, *Figure 3—figure supplement 2B–C*), or Lifeact::Ruby (*Figure 1E*, *Figure 7C*. *Figure 7—figure supplement 1C*). For mask generation, images were subjected to a gaussian blur filter and thresholded by intensity. Blur radius and the specific threshold algorithms used were empirically optimized for each experiment to consistently and accurately reflect the presynaptic area in control and mutant groups (and the same settings were used for all NMJs within any given experiment). Signal intensities were measured in 3D using a FIJI script, and colocalization analysis was performed in 3D on Airyscan or SIM reconstructed image stacks using the Coloc2 plugin for ImageJ (https://imagej.net/Coloc_2). For all images, background was subtracted using the rolling ball method with a radius of 50 pixels.

## In vitro assays
### Protein purification

His-Dap160 fragments were amplified from Dap160 isoform A and cloned into pTrcHisA (see 'Key resources' table for primer details). N-terminally His-Xpress–tagged proteins (Nwk[1-633], Nwk[1-731], Nwk[607-731], Nwk[1-428], Wsp[143-529], Dap160[SH3C], Dap160[SH3CD]) were purified as described previously (*Becalska et al., 2013*; *Kelley et al., 2015*; *Rodal et al., 2008*; *Stanishneva-Konovalova et al., 2016*). In brief, proteins were purified from BL21(DE3) *Escherichia coli* using cobalt or nickel columns, followed by ion exchange and gel filtration into 20 mM Tris, pH 7.5, 50 mM KCl, 0.1 mM ethylenediaminetetraacetic acid, and 0.5 mM dithiothreitol (DTT). GST fusions (Dap160[SH3CD], Dap160[SH3C], Dap160[SH3D]) were amplified from Dap160 isoform A and cloned into pGEX4t (see 'Key resources' table for primer details). Proteins were purified with glutathione agarose (Thermo Scientific, Waltham, MA) in 20 mM Tris 7.5, 20 mM KCl, and 0.5 mM DTT supplemented with protease inhibitors (P2714 [Sigma-Aldrich, St Louis, MO] and 0.5 mg/ml pepstatin A). Arp2/3 complex was purchased from Cytoskeleton, Inc. Actin was purified from acetone powder (*Spudich and Watt, 1971*) generated from frozen ground hind leg muscle tissue of young rabbits (PelFreez, Rogers, AR).

## Coprecipitation assays

Coprecipitation with GST-tagged proteins was conducted as described previously (*Kelley et al., 2015*). Concentrations of GST fusions on beads were normalized using empty beads and bead volume was restricted to two-thirds of the total reaction volume. GST fusions were incubated by agitation with His-tagged target proteins at room temperature for 1 hr in binding buffer (20 mM Tris, pH 8.0, 20 mM KCl, 0.5 mM DTT). For salt sensitivity experiments, the indicated concentrations of NaCl were used in place of KCl in the binding buffer. Beads were then pelleted and washed once with buffer after removing the supernatant. Pellets and supernatants were then boiled in Laemmli sample buffer and fractionated by SDS-PAGE, followed by Coomassie staining or immunoblotting as noted in figure legends, followed by imaging and analysis on a LICOR Odyssey device.

## Liposome cosedimentation

Lipid cosedimentation assays were conducted as described previously (*Becalska et al., 2013*). In brief, liposomes were swelled from dried lipid films in 20 mM 4-(2-hydroxyethyl)-1-piperazineethanesulfonic acid (HEPES), pH 7.5, and 100 mM NaCl. Specific lipid compositions are indicated in the figure legends. Proteins were then mixed with 1 mg/ml liposomes, incubated for 30 min at room temperature, and then pelleted for 20 min at 18,000 $\times g$ at 4°C. Pellets and supernatants were then denatured in Laemmli sample buffer and fractionated by SDS-PAGE, followed by Coomassie staining, and imaging and analysis conducted on a LICOR Odyssey device.

## GUV decoration

GUVs were generated by gentle hydration. Briefly, 10 µl of 10 mg/ml lipids dissolved in 19:1 chloroform:methanol were dried under vacuum, and then swelled in 300 µl of 5 mM HEPES 300 mM sucrose, pH 7.5, overnight at 70°C. GUVs were imaged on a Marianas spinning disc confocal system (see above). 3 µl GUVs were diluted into 5 mM HEPES 150 mM KCl, pH7.5, incubated with protein as noted in figures, and imaged using a x100/NA 1.4 objective in multiwell slides (Lab-Tek) precoated with 1 mg/ml bovine serum albumin. After 30 min of incubation, 1% agarose in 5 mM HEPES 150 mM KCl, pH 7.5, was added (final agarose concentration, 0.5%) to limit GUV mobility. Images were analyzed by line tracing intensity profiles across a medial optical section of GUVs.

## Actin assembly in droplets

Lipids (97.5% DPHPC [1,2-diphytanoyl-sn-glycero-3-phosphocholine] [Avanti Polar Lipids] and 2.5% DPHPC:PI(4,5)P$_2$) were mixed in chloroform, dried under vacuum, and rehydrated to 23 mM (20 mg/ml) in decane. The indicated proteins were added to the lipid mix at 1:50 vol ratio and pipetted vigorously until cloudy before imaging by spinning disc confocal microscopy.

## Pyrene-actin assembly

Rabbit muscle actin [5% (mol/mol) pyrene-labeled] was gel-filtered, prespun at 90,000 x$g$, exchanged from Ca$^{2+}$ to Mg$^{2+}$, and assembled at a final concentration of 2.5 µM as described previously (*Moseley et al., 2006*). Proteins were preincubated with 74 µg/ml liposomes or control buffer for 30 min before actin assembly reactions. Assembly was monitored with a spectrofluorometer (Photon Technology International) using an excitation wavelength of 365 nm and an emission wavelength of 407 nm. Rates were calculated from slopes of curves in the linear range, and curves were plotted using GraphPad Prism software.

## Quantification and statistical analysis

Graphs were prepared and statistical analyses performed using Graphpad Prism software. For normally distributed data, comparisons were made using either t-test or analysis of variance with posthoc Bonferroni's multiple comparisons test. For non-normally distributed data, comparisons were made using either Mann-Whitney U test or Kruskal-Wallis test with posthoc Dunn's test. No specific power analyses were performed; sample sizes were chosen based on established protocols and statistical analyses for significance, as detailed for all experiments here and in *Supplementary file 1*. Comparison of patch-duration distributions was performed using a Kolmogorov-Smirnoff test. Please see *Supplementary file 1* for each statistical test performed for each experiment presented in this study. All data are shown as the mean ± sem. Statistical significance denoted in all graphs *p<0.05, **p<0.01, ***p<0.001.

## Acknowledgements

The authors would like to thank Bruce Goode for actin reagents and advice, Julien Berro for particle tracking advice, Troy Littleton and Oleg Shupliakov for helpful discussions, Graeme Davis for anti-Dap160 antibody, and the Bloomington Drosophila Stock Center (Indiana University, Bloomington, IN, NIH P40OD018537) for providing fly stocks. This work was supported by a Basil O'Connor Scholar Award from the March of Dimes and a Pew Scholar award (AAR); by R01 NS116375 (AAR and TGF); by the Brandeis NSF MRSEC, Bioinspired Soft Materials (NSF-DMR 2011846); by T32 NS007292 (SJD), T32 GM007122 (CFK), and R90 DA03243501 (MFM); and by the Swiss National Science Foundation (grant 310030B_182825) and NCCR Chemical Biology funded by the SNSF (MK, MM).

# Additional information

## Funding

| Funder | Grant reference number | Author |
| --- | --- | --- |
| March of Dimes Foundation | | Avital Adah Rodal |

| Pew Charitable Trusts | | Avital Adah Rodal |
|---|---|---|
| National Institutes of Health | NS116375 | Thomas G Fai<br>Avital Adah Rodal |
| National Science Foundation | NSF-DMR 2011846 | Steven J Del Signore<br>Thomas G Fai<br>Avital Adah Rodal |
| National Institutes of Health | GM007122 | Charlotte F Kelley |
| National Institutes of Health | NS007292 | Steven J Del Signore |
| National Institutes of Health | DA032435 | Michelle F Marchan |
| Swiss National Science Foundation | 310030B_182825 | Markus Mund<br>Marko Kaksonen |
| Swiss National Science Foundation | NCCR Chemical Biology | Markus Mund<br>Marko Kaksonen |

The funders had no role in study design, data collection and interpretation, or the decision to submit the work for publication.

### Author contributions

Steven J Del Signore, Conceptualization, Data curation, Software, Formal analysis, Validation, Investigation, Visualization, Methodology, Writing - original draft, Writing - review and editing; Charlotte F Kelley, Conceptualization, Resources, Data curation, Formal analysis, Investigation, Visualization, Methodology, Writing - original draft, Writing - review and editing; Emily M Messelaar, Conceptualization, Resources, Formal analysis, Investigation, Methodology; Tania Lemos, Michelle F Marchan, Investigation; Biljana Ermanoska, Investigation, Writing - review and editing; Markus Mund, Resources, Writing - review and editing; Thomas G Fai, Methodology; Marko Kaksonen, Resources, Supervision, Funding acquisition; Avital Adah Rodal, Conceptualization, Data curation, Supervision, Funding acquisition, Validation, Visualization, Methodology, Writing - original draft, Project administration, Writing - review and editing

### Author ORCIDs

Steven J Del Signore (iD) https://orcid.org/0000-0001-6007-9732
Charlotte F Kelley (iD) http://orcid.org/0000-0002-7684-9049
Tania Lemos (iD) http://orcid.org/0000-0002-6710-7676
Markus Mund (iD) http://orcid.org/0000-0001-6449-743X
Thomas G Fai (iD) http://orcid.org/0000-0003-0383-5217
Marko Kaksonen (iD) http://orcid.org/0000-0003-3645-7689
Avital Adah Rodal (iD) https://orcid.org/0000-0002-2051-8304

### Decision letter and Author response

Decision letter https://doi.org/10.7554/eLife.69597.sa1
Author response https://doi.org/10.7554/eLife.69597.sa2

## Additional files

### Supplementary files

• Source code 1. This file contains the code used in this manuscript as follows. Code used to generate toy data analyzed in *Figure 6—figure supplement 2*. Code used to analyze toy data analyzed in *Figure 6—figure supplement 2*. Code used to analyze channel intensity and distribution for *Figure 1—figure supplement 1F*, *Figure 3D*, *Figure 3—figure supplement 2B-D*, *Figure 7F–I*, *Figure 7—figure supplement 3*. Code used to analyze channel colocalization in 3D for *Figure 1F*, *Figure 2D,F*, *Figure 3E*, *Figure 7C*, *Figure 7—figure supplement 1C*.

• Supplementary file 1. Summary of genotypes and statistics for all experiments in this study.

• Transparent reporting form

## Data availability

Source data files and source code have been provided for all figures accompanying this manuscript.

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
