## [Decision Letter]

[Editors' note: this paper was reviewed by Review Commons.]

**Acceptance summary:**

Live-cell microscopy of *Drosophila* larvae shows that actin patches assemble transiently in unstimulated neuromuscular junctions at the frequency of spontaneous synaptic vesicle release. Disruption of the interactions between Dap160 and Nwk in vivo leads to dysregulated and more frequent actin assembly. The authors propose that these interactions serve as a 'clamp' on the actin assembly machinery, so that it can be spatially and temporally controlled when needed for synaptic vesicle endocytosis.

---

## [Author Response]

We thank the reviewers for their constructive comments. We are pleased that reviewers:

1) Found our model for an autoinhibitory clamp to be an important and convincing advance in our understanding of synaptic actin assembly and function.

2) Considered our in vitro data to be rigorous and convincing, and our in vivo analysis of actin dynamics ‘an impressive feat’

3) Were convinced of the mechanism and significance of Nwk-Dap160-WASp in the regulation of actin assembly at synaptic membranes.

The primary question raised by reviewers was the extent to which actin is shown to be endocytic. We addressed this concern by more extensive and rigorous imaging and analysis of the relationship between endocytic adapters and actin patch dynamics (new Figure 7A-C, Figure 7—figure supplement 1A; and response to comments R1S, R1.4, R1.11, and R2.1). Other comments to be addressed by revisions are discussed point-by-point below.

Summary of revision experiments

Experiment 1 – Compare Clathrin and Actin dynamics

We have now added a new analysis of clathrin::GFP relative to Lifeact::Ruby, and show kymographs and line traces of the dynamics between clathrin and lifeact patches (new Figure 7A-C).

Experiment 2 – Provide control for AP2 and Arp3 co-localization data

In response to comment R2.1, which requests a negative control for colocalization analysis of AP2-lifeact and Arp3-lifeact, we compared actin distribution with the active zone marker BRP (new Figure 1E-F and Figure 7C)

Experiment 3 – Validate patch-tracking parameters

In response to comment R2.10, we have validated our patch tracker analysis as follows:

We present a new analysis that compares the frequency of patch detection in control NMJs at 0.25 Hz and 1 Hz across a wide patch detection and tracking parameter space (new Figure 1—figure supplement 2A and Figure 6—figure supplement 1A).

We show that the phenotypes presented for WASp and Nwk in the original submission are highly robust with respect to the patch detection and tracking parameters we chose (new Figure 1—figure supplement 2B and Figure 6—figure supplement 1B).

In videos and representative panels we now label tracked patches, and show representative kymographs to clarify the dynamics of structures detected by patchtracker (new Figure 1—figure supplement 1A, Figure 7A-B, revised Video 1).

Experiment 4 -Validate WASp RNAi

In response to comment R2.13 regarding Figure 1—figure supplement 1, we quantify the efficiency of WASp knockdown by the RNAi construct used (new Figure 1—figure supplement 1F).

Reviewer #1 (Evidence, reproducibility and clarity (Required)):Summary:R1S. The authors provide extensive biochemical and in vivo evidence that together map interactions between Dap160 and Nwk (*Drosophila* homologues of intersectin and FCHSD2, respectively) that regulate the ability of Nwk to activate WASp and hence actin assembly at the synapse. They show in vitro that synergistic interactions between the SH3C domain of Dap160, the SH3b domain of Nwk and membrane-associated PI45,P¬2 relieves autoinhibition of Nwk to stimulate WASp and Arp2/3-mediated actin assembly. By live-cell microscopy of neuromuscular junctions (NMJs) in Ds. larvae, they show that actin patches assemble transiently in unstimulated NMJs at about the frequency of spontaneous synaptic vesicle release with lifetimes of ~20-40s. Disruption of the interactions between Dap160 and Nwk in vivo leads in dysregulated and more frequent actin assembly. Their hypothesis is that these interactions serve as a 'clamp' on the actin assembly machinery, so that it can be spatially and temporally controlled when needed for synaptic vesicle endocytosis. The data linking these observations to synaptic vesicle endocytosis are somewhat weaker, as most sites of AP2 assembly on the membrane are devoid of actin and disruption of Nwk or Nwk-Dap160 interactions results in a considerably milder endocytosis phenotype than Shits1 the mutation. However, the weaker phenotype may reflect other mechanisms operating to regulate endocytosis and actin assembly at the synapse. This should perhaps be discussed more explicitly.‘most sites of AP2 assembly on the membrane are devoid of actin’

Indeed, this is the main observation that sparked our study, and turns out to be a surprising and interesting feature of many components of the synaptic endocytic machinery. The wide distribution of AP2 and clathrin (though we note that we cannot make assumptions about their assembly state) is consistent with the high concentration (Wilhelm et al., 2014) and expansive localization of many endocytic regulators at synapses. The conundrum is that the expected rate of exo/endocytosis at this synapse at rest (Melom et al., 2013) is far too low to account for this broad distribution. Thus, these data argue that most of the endocytic machinery is held inactive at synaptic membranes, and is thus unlikely to always be associated with force-producing actin assembly.

The converse question is whether the actin-positive events are really functionally associated with a subset of the endocytic machinery. The following lines of evidence suggest that they are functionally associated:

1. Nwk and Dap160, which we find regulate actin in vitro and at patches at the synapse, are well-established regulators of endocytosis in other cell types.

2. We find that dynamin, a central component of the endocytic machinery, also regulates synaptic actin patch dynamics.

3. The rate of actin assembly matches the measured rate of exocytosis (and thus the expected rate of compensatory endocytosis)

4. Actin patch dynamics resemble endocytic patch dynamics in other cell types.

5. Actin/WASp is required for endocytosis in this cell type, and we find that WASp is required for formation of synaptic patches (though note we observe only a partial loss of patches since we have used a hypomorphic WASp mutant). Further, the patches are highly enriched for Arp2/3 complex, as expected for endocytic structures.

Added new experiment (Figure 7A-C): To further test the hypothesis that the actin patches are bona fide sites of endocytosis, we provide a more rigorous quantification of the relationship between Clc and actin and show tracked actin patches that dynamically colocalize with clc::GFP. Please see further discussion on this point in response to Comment 1.4.

‘disruption of Nwk or Nwk-Dap160 interactions results in a considerably milder endocytosis phenotype than Shits1 the mutation’

The magnitude of the Nwk-Dap160 FM dye uptake phenotype is similar to many other endocytic proteins at this synapse (Choudhury et al., 2016; Guichet et al., 2002). The simplest explanation for the more severe Shi^TS1^ phenotype is that it is a dominant negative mutant, completely blocking endocytosis by locking constricted pits, while the Nwk and Dap160 mutants are loss of function perturbations, allowing for redundant mechanisms that provide resiliency in the endocytic protein interaction network (Chen and Schmid, 2020). Indeed, even dynamin loss-of-function mutations cause more subtle phenotypes than the classic dominant negative mutants (Ferguson et al., 2007). There are dozens of endocytic proteins with overlapping molecular interactions and activities, which suggests several explanations for the intermediate Nwk/Dap160 phenotypes. First, it could reflect direct molecular redundancy with other adapters/regulators. Alternatively, it could reflect an adaptive change in the mode of membrane uptake in the mutants (as is observed in clathrin and adapter mutants, eg Heerssen et al., 2008). Regardless, the combination of actin and endocytic phenotypes of the Dap160^∆SH3CD^ mutant provides strong support that the actin regulatory function of the Nwk-Dap160 module are required for normal endocytosis.

Major comments:- Are the key conclusions convincing?R1.1 The biochemical studies are very convincing. The in vivo studies are supportive, but as indicated above not as strong, perhaps unavoidably.- Should the authors qualify some of their claims as preliminary or speculative, or remove them altogether?R1.2 The idea of an endocytic 'clamp' is attractive, and others have suggested that this might be accomplished through phosphorylation/dephosphorylation. Indeed, while the authors show clearly that Dap160-Nwk interactions regulate Nwk activity, they do not comment on what might regulate Dap160-Nwk interactions. What would trigger these interactions for activation of Nwk?

We have added additional text discussing possible modes of regulation of the Dap160-Nwk interactions. Phosphorylation is one possibility, but is perhaps more likely to provide the kind of regulation that switches between modes of endocytosis (eg CME to bulk) over time windows in the seconds-minutes range (Salazar and Höfer, 2009). However, it seems likely that within a single mode (eg CME), this would not be fast enough for release of the clamp in response to an action potential. A plausible sequence of events that couples exocytosis and endocytosis suggests several potential ‘switching’ mechanisms, which are very interesting but beyond the scope of this study to explore.

Sequence of events:

Pre-deployment of endocytic-actin regulatory machinery (‘clamped’)

Calcium influx (may directly act on the endocytic machinery to release the clamp)

Exocytosis may also/alternatively provide the signal to release the clamp via:

– Cargo accumulation in the plasma membrane (eg vesicular SNAREs).

– Local changes in membrane composition and charge: Phosphoinositide dynamics could directly alter Nwk membrane binding and/or protein-protein interactions. Alternatively, Dap160/Intersectin activity may be altered by interactions with other PI(4,5)P2 binding proteins, including AP2, FCHo or Dynamin. Finally, membrane composition changes upon exocytosis may directly recruit or activate WASp.

– Local changes in membrane mechanics or curvature

Added text lines 399-405: The next critical step will be to determine the mechanisms that control switching between clamped and activated states. Many potential mechanisms that link calcium-dependent exocytosis and endocytosis could activate actin assembly, including direct effects of calcium on the endocytic machinery (Maritzen and Haucke, 2018), the accumulation of synaptic vesicle cargoes (Cousin, 2017), stoichiometry-dependent changes in protein interactions or activities (Case et al., 2019), changes in membrane mechanics (Anantharam et al., 2010; Dai et al., 1997; Roux et al., 2010), and changes in membrane charge/mode of membrane binding (Kelley et al., 2015).

- Would additional experiments be essential to support the claims of the paper? Request additional experiments only where necessary for the paper as it is, and do not ask authors to open new lines of experimentation.R1.3 Given the partial effect on endocytosis upon disruption of Dap160/Nwk and the fact that most AP2 puncta do not colocalize with the actin patches, I'm curious as to the effect of latrunculin A on FM dye uptake. Does latA inhibit as strongly as shits1? I believe this is a fairly do-able experiment.

This experiment has already been published, and supports our model. Latrunculin A treatment strongly inhibits FM dye uptake at the *Drosophila* NMJ (Wang et al. 2010). Though this study did not compare the degree of defect to shiTS1, they report a ~80% decrease, which is very similar to our reported effect of shiTS1 (73% decrease). We cite this result (p2) as one piece of evidence to support the premise that endocytosis in this synapse is actin dependent.

R1.4 Or perhaps the AP2 puncta are not endocytic events. The authors state they are more dynamic than the actin patches, what is their lifetimes/distribution of duration relative to actin patches.

As mentioned above, we do believe that most AP2 puncta are unlikely to be active sites of endocytosis. If all puncta were active, endocytosis would far exceed exocytosis at this synapse. These abundant AP2 puncta may be in small clathrin pre-assemblies, or (less likely) serve some non-endocytic function. This is true also for the broad membrane localization of the other endocytic proteins analyzed here and in other work, which together suggest that mechanisms must be in place to limit the activity of these proteins. Indeed, this is the very observation that animates the question we are asking – how do synapses maintain such a high (inactive) concentration of membrane and cytoskeletal regulators at synaptic membranes?

Added text lines 269-271: Considering that the rates of exo/endocytosis at this synapse at rest are relatively low (see above), these observations suggest that like other periactive zone endocytic proteins, a large pool of membrane localized clathrin coat and adapter proteins are not actively engaged in endocytosis.

Added text lines 356-360: This is further consistent with the broad distribution and transient localization of AP2 to the membrane (Figure 7). Given the comparatively low rate of endocytosis expected at rest at this synapse, this suggests that most AP2 puncta at the synapse are either not stabilized to form productive endocytic sites (Aguet et al., 2013), or are associated with some non-endocytic function (Gimber et al., 2015).

Added new experiment (Figure 7A-C) as described above. A deeper analysis of the co-dynamics and functions of particular actin-clathrin structures will be interesting but very complex given the challenges of analyzing these dynamics at the synapse as compared to non-neuronal membranes, and we believe more appropriate as the topic of a future manuscript.

R1.5 In this regard, while the actin patch lifetime is indeed consistent with previous reports in yeast and cultured mammalian cells, I would think that synaptic vesicle endocytosis occurs at a much more rapid time scale, so I'm not sure of the validity of this comparison.

Vesicle endocytosis occurs over a range of time scales at this and other synapses, depending on the degree of exocytosis (Gan and Watanabe, 2018). Endocytosis at this synapse proceeds reliably with a time constant of ~14 sec with 50Hz stimulation over a range of durations, consistent with clathrin mediated endocytosis (Poskanzer et al., 2006), and multiple studies have shown that normal vesicle cycling required clathrin and adapters. These data lead us to expect similar actin dynamics between these model systems. We note also that actin patch lifetimes differ significantly between yeast (~16sec) and cultured mammalian cells (~40sec), with our measurements closely matching yeast patch lifetimes. We have revised the text to clarify this argument:

Added text lines 331-334: Given the measured time constant for endocytosis (~14 seconds, Poskanzer et al., 2006) and clathrin-dependence of vesicle cycling in this synapse (Heerssen et al., 2008), these values support the hypothesis that synaptic actin patches are likely sites of clathrin-mediated endocytosis.

- Are the data and the methods presented in such a way that they can be reproduced?Yes.- Are the experiments adequately replicated and statistical analysis adequate?Yes.Minor comments:• R1.6 Page 2, left column. What is meant by 'we noted a strong floor effect'.

We mean that the mode of the distribution was the minimum observable duration, suggesting that we did not adequately resolve the population of shorter-lived events. We clarify this point in the text (p2):

Added text line 110-111: We did note a high percentage of patches in the minimum duration bin, suggesting the existence of even briefer patches.

• R1.7 Figure 5F. The fluorescence recovery curve for the control condition shows biphasic kinetics, which would be expected as there is both a membrane-bound and a cytosolic pool of Nwk:GFP. Tau should be recalculated. I suspect the fast phase will be the same as for Dap160siRNA and the slow phase will show a much bigger difference.

We now fit recovery data to two-phase curves, and indeed found that the fast component showed similar kinetics (.75 sec vs.74 sec), and statistically indistinguishable slow kinetics (10.7 sec vs 16.4 sec; p=.08). Strikingly, fast kinetics accounted for a significantly higher fraction of recovery in dap160RNAi boutons (76.2% vs 22.2%, p<.05). These data strongly support a shift from membrane to cytosolic localization and dynamics. While our conclusion remains the same, we agree with the reviewer that this is the more appropriate analysis of the data and have revised figure 5.

• R1.8 Page 10, left column. It is surprising that Dap160 mutants can rescue synaptic growth regulation. Is this process independent of endocytosis. Does shits1 block synaptic growth regulation?

It was also surprising to us that Dap160^∆SH3CD^ could rescue synaptic growth, though *nwk* null and *dap160* null mutants have synaptic growth phenotypes of varying degrees, which arise from misregulation of BMP signaling. Multiple other endocytic mutants, including Shi^TS1^, show the same phenotype (Dickman et al., 2006; O’Connor-Giles et al., 2008), so we do not believe that it reflects a non-endocytic process.

This surprising result with Dap160^∆SH3CD^ is interesting for two reasons:

1. it indicates for the first time that endocytic mechanisms for synaptic vesicles/FM dye can be uncoupled from those required for growth factor receptor trafficking.

2. It raises the possibility that these do occur by somewhat different mechanisms. One interpretation is that the mechanism reported by FM dyes may be simply more sensitive to the Dap160-SH3CD perturbation than synaptic growth. Another possibility is that other ligands can activate Nwk for growth factor receptor trafficking, and that the Dap160-SH3CD mechanism is dedicated to synaptic vesicle endocytosis, perhaps as a rapid response to action potentials. Regardless, the multivalent nature of interactions in the endocytic network (Dap160SH3AB-Dynamin, Dynamin-Nwk, Dap160SH3CD-Nwk, Dap160SH3AB-WASp, WASp-NwkSH3a, just to name a few) will make this challenging to tease apart.

Added text lines 283-285: These data indicate that synaptic vesicle and growth factor endocytosis are mechanistically separable, and suggest that actin dynamics phenotypes in the Dap160^ΔSH3CD^ mutant are not associated with synaptic growth regulation.

• R1.9 Page 10, right column. It's confusing to report 28% and 27% reduction for Nwk and DAP160 mutants and 28% of controls for Shits1. This should be reported also be reported as 72% reduction to be consistent.

We agree this is confusing and have fixed this in the manuscript.

• R1.10 Is it clear that Dap160, Nwk and actin correspond to "much of the synaptic membrane remodeling machinery" as stated on pg 11, right column? What about other BAR domain-containing proteins (endophilin, amphiphysin, syndapin) and dynamin?

The conclusion that much of the machinery is broadly deployed is based both on our observations of Nwk, Dap160, and AP2, and published reports for dynamin (Roos and Kelly, 1998), Synaptojanin (Verstreken et al., 2003), Eps15 (Koh et al., 2007), and EndophilinA (Guichet et al., 2002; Verstreken et al., 2002). We have clarified the basis for this conclusion in the text on p11.

We note here for completeness that some endocytic proteins, such as Syndapin (Kumar et al., 2009) and amphiphysin (Leventis et al., 2001; Razzaq et al., 2001; Zelhof et al., 2001) are undetectable presynaptically and not thought to function in presynaptic endocytosis at the fly larval NMJ.

Added text lines 350-354: Many endocytic proteins accumulate across broad membrane domains at the *Drosophila* NMJ and other synapses (Gerth et al., 2017; Guichet et al., 2002; Koh et al., 2007; Roos and Kelly, 1998; Verstreken et al., 2002, 2003). Our data indicate that much of this membrane remodeling machinery is likely held in an inactive state at the presynaptic membrane: Nwk and Dap160 accumulate in a micron-scale membrane domain (Figure 2), and their loss increases the frequency of short-lived actin patches (Figure 6).

• R1.11 Same page, while it is true that "a significant and measurable fraction of synaptic actin patches are associated with endocytosis" as stated, the converse appears not to be true (i.e. most of the endocytic sites marked by AP2 are not associated with actin). As discussed above this should be clarified.

As discussed above, it is precisely the high frequency of AP2 at the membrane, combined with the high membrane concentration of cytoskeletal regulators such as Nwk and Dap160 that prompted us to ask whether there is a mechanism to constrain actin assembly at the synapse. Please see discussion for R1.4, and see Revision Plan Experiment 1 for further discussion.

Added text lines 356-360: This is further consistent with the broad distribution and transient localization of AP2 to the membrane (Figure 7). Given the comparatively low rate of endocytosis expected at rest at this synapse, this suggests that most AP2 puncta at the synapse are either not stabilized to form productive endocytic sites (Aguet et al., 2013), or are associated with some non-endocytic function (Gimber et al., 2015).

• R1.12 The authors might wish to speculate on how Dap160-Nwk interactions are regulated in a spatial temporal manner and are they?

Added text lines 399-405: The next critical step will be to determine the mechanisms that control switching between clamped and activated states. Many potential mechanisms that link calcium-dependent exocytosis and endocytosis could activate actin assembly, including direct effects of calcium on the endocytic machinery (Maritzen and Haucke, 2018), the accumulation of synaptic vesicle cargoes (Cousin, 2017), stoichiometry-dependent changes in protein interactions or activities (Case et al., 2019), changes in membrane mechanics (Anantharam et al., 2010; Dai et al., 1997; Roux et al., 2010), and changes in membrane charge/mode of membrane binding (Kelley et al., 2015).

R1.13 These studies have examined actin patch dynamics only in unstimulated NMJ. Do actin patches form more frequently and perhaps associate more frequently with AP2 upon stimulation?

Our study focuses on unstimulated presynaptic terminals because we are primarily interested in the mechanisms that maintain the endocytic machinery at the membrane in a quiescent state. We agree that the effect of stimulation on the activity of the endocytic machinery is interesting, however this technically challenging experiment has a wide range of possible outcomes compatible with roles for actin in endocytosis. These include no change in perceived actin dynamics (if the amount of membrane uptake scales independent of actin rate/size), increased frequency of assembly events, increased amplitude of assembly events, or more unpredictable changes if the mode of uptake switches altogether with stimulation (e.g. to bulk uptake). Thus, we believe that the amount of work required to interpret any result from this line of questioning would be beyond the scope of this manuscript.

Reviewer #1 (Significance (Required)):- Describe the nature and significance of the advance (e.g. conceptual, technical, clnical) for the field.These studies provide insight into a potential mechanism for the spatial and temporal regulation of actin dynamics and hence endocytosis at neuromuscular junctions. The combination of live imaging, genetic perturbations and biochemical analyses make for a complete and rigorous story. The hypothesis for such a molecular 'clamp' is attractive and supported by the data.- Place the work in the context of the existing literature (provide references, where appropriate).The authors do a good job of this. To my knowledge these are the first studies showing high resolution imaging of individual actin patch assemblies at a NMJ and, while the authors had previously mapped Dap160-Nwk interactions the functional consequences with regard to Nwk regulation we not explored.- State what audience might be interested in and influenced by the reported findings.The studies will be of interest to those studying endocytic membrane trafficking both at the synapse and in general.- Define your field of expertise with a few keywords to help the authors contextualize your point of view.Endocytic membrane trafficking.Reviewer #2 (Evidence, reproducibility and clarity (Required)):This manuscript examines the role of endocytic proteins Nervous Wreck (Nwk), Intersectin (Dap160 in *Drosophila*) and WASp in actin dynamics associated with endocytosis in vitro and in vivo in the neuromuscular junction (NMJ) of *Drosophila larvae*. The authors quantify individual F-actin assemblies by spinning disk confocal in the NMJs presynaptically expressing three different fluorescent actin probes, and compared the actin patch dynamics at 0.25 Hz and 1 Hz acquisition rates using automated particle tracking and quantifications tools. Using these tools, authors identified predominantly transient actin patches, and then go on to demonstrate (using genomic and RNAi manipulations) that these actin structures required WASp-dependent activation of Arp2/3. They then investigated the role of WASp regulators NwK and Dap160 using conventional and super-resolution microscopy at the periactive zone (PAZ). They showed that actin patches were more sparsely dispersed than Nwk and Dap160 in respect to PAZ, suggesting that PAZ machinery may be locally self-regulated to control actin patches. The authors then demonstrate that Nwk autoinhibition occurs through multiple interactions between Nwk and Dap160, and that these interactions are involved in synaptic endocytosis. The author conclude that this autoinhibition clamps and primes the synaptic endocytic machinery to drive productive membrane remodelling in response to physical cues. Overall, the role of WASp, Nwk and Dap160 on membrane-associated actin dynamics in presynaptic endocytosis is an important discovery. However, there are a number of issues with the current manuscript.Major comments:R2.1 The first part of the manuscript is convincing demonstrating the role of WASp, Nwk and Dap160 in actin patch dynamics. The experiments showing that these actin structures are involved in synaptic endocytosis would need some more work. The authors show that Lifeact::Ruby co-localize with AP2::GFP (Figure 7A-B, Video 3). They should show a negative presynaptic control that does not localize with Lifeact::Ruby.

Added new experiment (Figure 1F, Figure 7C): To address this question, we have compared the accumulation of lifeact::Ruby with the active zone marker BRP::GFP as a negative control and repeated our analysis of Arp3::GFP-Lifeact colocalization, and added a new comparison between Lifeact::Ruby and presynaptically expressed clc::GFP. We believe this is a rigorous experimental negative control because (i) the vast majority of BRP signal is localized to the membrane (ii) BRP localizes to a dense array of punctate structures morphologically similar to AP2/clc/arp3 puncta (iii) BRP exhibits a highly complementary pattern of accumulation with endocytic proteins such as AP2. As expected, BRP exhibits an exceedingly low Pearson’s Correlation with Lifeact::Ruby (likely reflecting background levels of colocalization at the resolution of light microscopy due to their general membrane localization). Both Arp3 and Clathrin light chain exhibit significantly higher colocalization with Lifeact::Ruby, supporting the conclusion that these relationships are not spurious.

R2.2 The Video 3 is hard to follow, perhaps it would help if the authors would indicate with arrows what they like to point out, or show a smaller region of interest.

We will have replaced Video 3 with a new video, and marked patches of interest with arrows.

R2.3 Of note, the AP2::GFP signal appears very abundant, which makes the use of a negative control even more important.

Agree, see discussion to R1.4.

R2.4 The authors show that dominant negative dynamin shiTS1 mutants have impaired FM dye uptake, and decreased number of actin patches, and draw conclusion that normal actin patch assembly requires dynamin activity. It has been shown before that dynamin is part of a protein network that controls nucleation of actin from membranes and that Nwk interacts with dynamin and Dap160 and functions together with Cdc42 to promote WASp-mediated actin polymerization in vitro and to regulate synaptic growth in vivo (Rodal et al., 2008). The manuscript would perhaps be stronger if the authors would address the combined role of the proteins forming the autoinhibitory clamp, and actin polymerization promoting protein dynamin. Otherwise the dynamin results provide more of a stand-alone observation.

We used the Shi^TS1^ dynamin mutant as a well-characterized and robust tool to block membrane internalization at the synapse (in fact more severe than a typical loss of function mutant in the endocytic machinery, due to the dominant negative effect of Shi^TS1^ in arresting invaginating pits, see response to R1 overall comments). While we certainly agree that further investigating the multiple roles of dynamin in this process (through its membrane scission activity, its own actin association, and its interaction with ligands of its proline-rich region such as Nwk and Dap160-SH3AB) would be very interesting, it will require multiple years and several additional papers worth of work to sort these out. We do feel that the biochemical mechanisms we have isolated in vitro for Nwk, WASp, and Dap160, and tested in vivo with specific mutants in Dap160 that are not involved in dynamin binding, stand alone as a first step in this much longer journey.

R2.5 Table 1 indicates several experiments with 3 technical replicas per lane. The authors should specify what these technical replicas mean. Were these experiments performed once and the samples loaded 3 times, or are these 3 independent repetitions of each experiment?

These indicate three independent repetitions/reaction assemblies, and we have modified the text to clarify.

R2.6 The statistical analysis are, in general, robust and the Table 1 listing the used statistical tests is very useful. The authors should include negative controls and statistical analysis for all stand-alone experiments such as Mander's M1 in Figure 1F and 7C.

Agreed, see discussion of point R2.1

Minor comments:R2.7 Prior relevant studies have been appropriately referenced, the figures are clear and appropriately pointed out in the text (with the exception of Page 10, where FM uptake experiments are written [Figure 7A-B], when it should be [Figure 7F-G]). The main text is well written and understandable.Specific experimental issues are listed below:R2.8 In general, there is a large variation of the NMJ area (µm2) (for example, Figure 1A, 3D, 5E, 1—figure supplement 1A and 1—figure supplement 1C). The authors should clarify which muscle NMJs were used in each experiment. What is the average NJM size? Does the number of the actin patches correlate with the presynaptic size, i.e. are different actin structures more abundant in larger presynapses? For example, the actin cytoskeleton and its regulatory proteins are different in dendritic spines that are at different maturity stage (i.e. filopodia, stub, mushroom. For reference please see for example Hlushchenko et al., 2016 Cytoskeleton (Hoboken) 73(9):435-41. doi: 10.1002/cm.21280). Are the different NMJ presynapses equally active?

We modified the text to better describe the NMJ system. We performed our experiments on wandering third instar larvae, at type Ib boutons on muscle 6/7 NMJs, in abdominal segments 3-4. This synapse typically contains ~80 boutons stretched over ~200 µm of muscle, with a stereotypical distribution of bouton size (~6-16um2, Lee et al., 2017; Mallik et al., 2017) and a typical active zone density (~1/um2, so ~6-16 active zones/bouton; Harris and Littleton, 2015; Saburova et al., 2017). We have imaged a sub-area of this synapse, entailing a string of 5-10 boutons from each NMJ. We have then normalized all of our actin patch measurements to the synaptic area in each image. We find that both active zone and actin patch density are similar across the typical bouton size distribution for this NMJ. We note that “synaptic strength” at the NMJ is determined by individual active zone maturity and release probability, not by the size of the bouton (in which 6-16 independent, non-motile active zones reside) (Akbergenova et al., 2018).

Overall, the effects we see are not likely to reflect different stages of neuronal maturity from image to image (because we perform all experiments at late third instar), or spatial variation in synapse physiology (because release probability is distributed similarly between active zones along the length of the NMJ (Akbergenova et al., 2018)).

Thus, in short answer to the reviewer’s questions, all measurements are already normalized to NMJ area, and different boutons are indeed equally active when summed across their multiple active zones.

Added text lines 90-93: To control for developmental variation, all experiments were performed on late third instar larvae (~96-120 hours after egg laying) on muscle 6/7 NMJs at abdominal segments 3-4. The development and physiology of these synapses is well characterized (Harris and Littleton, 2015).

Added text p2: We normalized patch frequencies by the synapse area measured, and present data per 10 µm2, which is approximately the size of a synaptic bouton in this system.

R2.9 Number of actin patches (Figure 1 and Figure 1—figure supplement 1). The authors calculate that at 0.25Hz (i.e. 1 frame/ 4 sec),1.2 GMA patches/10μm2/min were observed (Figure 1B-D). In Figure 1—figure supplement 1D, the number of patches is related to NMJ area/Time (please define what "Time" is). Would the control results shown in Figure 1—figure supplement 1D be similar to those shown in Figure 1C for GMA if the number of patches was related to 10 µm2/min? Would it make more sense to show the number actin patches per average NMJ size?

We apologize for the inconsistent and confusing labeling of axes. We have fixed all axis labels to fully indicate units. All data shown are normalized in the same way, to 10 um2/min (and are thus already normalized to NMJ size). The reason we picked the value of 10 µm2 is because that represents the size of a typical bouton, and is therefore intuitive relative to the image. Given that the measurements are all done the same way, there is not a significant difference between control values for Figure 1 and Figure 1—figure supplement 1.

R2.10 Patch detection for the analysis of actin dynamics. Video 1 shows actin patches labelled by complementary reporters. Based on the Video 1, the size of actin patches in Actin::GFP is larger than those shown in GMA or Lifeact::Ruby videos. The patch diameter was selected as 0.6 µm for the analysis of actin dynamics at the NMJ (based on Methods), which appears correct for Actin::GFP but it appears it may be an overestimate for GMA or Lifeact::Ruby (there appears to be a large number of smaller actin patches especially in Lifeact::Ruby video). The smaller patches appear to be more short lived than the bigger patches, especially in Actin::GFP video. Some of the actin patches also appear laterally mobile. The linking max distance 0.5 µm appears large considering the 1 frame / 4s acquisition frame rate, which may results in significant tracking defects. The authors should show evidence that the 0.25 Hz acquisition rate, which has been used in majority of the experiments, with linking max distance 0.5 µm, is adequate enough to detect and discriminate this type of mobility and that there is not major mistracing of the actin separate patches. The Actin::GFP signal is very high (close to saturated) at the beginning of the video: how are the individual patches detected, tracked and counted reliably? The authors should show evidence how the patches were identified and tracked by the software in time-lapse. Please also specify what the time scale in the upper right corner of the different videos is. Could you also please show kymographs of the actin live cell imaging, correlating the patch duration with the length of the kymograph signal?

The submitted videos were contrast adjusted so that the reader could see both bright and dim structures; please be assured that all of our data were collected within the linear range of our detector; none of the raw data from which measurements were made were saturated.

R2.11 Figure 1E: Does the Arp3 and lifeact signal signal originate from presynapses or adjacent axons?

We do not observe dynamic, bright actin structures in the bon-fide axonal region at the nerve entry point into the muscle. The interbouton regions at these synapses represent an exceedingly small fraction of the measured area; while we do not exclude these regions from our analysis, they are unlikely to significantly contribute to our measurements.

R2.12 Figure 6A-C and 1—figure supplement 1A: The authors point out actin cables (magenta) and dynamic actin patches (green arrowhead) in Figure 1—figure supplement 1A, and quantify the patch duration in Figure 6C. Based on Figure 1—figure supplement 1A, it appears that great majority of actin patches, which are not indicated in the figure, are stable and only a few patches disappear within the indicated 32 s time frame along with the one indicated with green arrowhead. Yet the frequency distribution in Figure 6C indicates that the great majority of the control patches are short-lived (0-10s). The authors should explain the discrepancy.

Our method analyzes only patches that begin and end within the imaging window. This certainly excludes actin structures of a very long-lived nature (such as those likely to be involved more directly in bouton growth; Piccioli and Littleton, 2014). We discuss the caveats and constraints on this quantification in the results and methods sections of the text, and emphasize that our measurements reflect the properties of the subset puncta that are dynamic over the duration of videos.

R2.13 Figure 1—figure supplement 1 C-E: The authors should show the KD efficiency of WASp RNAi. Please also note varying labelling WASp (is it WASp or wsp?).

We have revised the manuscript to make labels consistent.

New experiment (Figure 1—figure supplement 1F): We analyzed the degree of knockdown of a presynaptic WASp::myc transgene by the same RNAi construct used in Figure 1—figure supplement 1C-D. We find that WASp accumulation is reduced to background levels. We chose to quantify a presynaptically expressed transgene due to the fact that the majority of endogenous WASp labeling at the NMJ is postsynaptic. Given the limits of resolution of light microscopy and the close apposition of pre and postsynaptic membranes, this precludes quantitative analysis of knockdown of the endogenous protein, but we believe the transgene makes a good proxy to evaluate the effectiveness of the RNAi line.

R2.14 Figure 2: It is unclear what the merged channels are in the small insets in A, C and E. Please clarify. What is the scale bar in A, C and E?

The channels in the insets are pseudo-colored as in the larger panel, as labeled. We have re-colored the fonts in the single-channel insets to emphasize this.

R2.15 Figure 3: What is the scale bar in E? The Mean Nwk Intensity of Dap160FL and Dap160ΔSH3D in D is so high that it basically covers the who NMJs, which would result in high colocalization in E. The Dap160 ΔSH3CD rescue expression level in E appears significantly lower compared to the other Dap160 variants – could this be the reason why the Nwk Mean Intensity is significantly lower than in the other two Dap160 variants (graph in D)?

The scale bar in D is 5um, the scale bar in E is 2.5um. We have added this information to the legend.

We present a new replicate of the experiment presented in Figure 3E and quantify transgene expression in Figure 3 —figure supplement 2B. This experiment precisely replicates the original finding, though we did find that the levels of the Dap160^ΔSH3D^ transgene were lower in this specific experiment. However, this did not result in a loss of Nwk expression nor colocalization and so we conclude the levels of expression were sufficient for normal function. Regardless, the primary conclusion from this experiment is that Dap160^ΔSH3CD^ diminishes its colocalization with Nwk, and this transgene is expressed at indistinguishable levels from the full length control transgene. We have chosen new representative images and have specified the transgene expression in the text.

New text lines 170-172: Notably, truncation of Dap160SH3D did not exhibit a phenotype in these assays despite lower levels of expression in this assay (Figure 3—figure supplement 2B), suggesting that additional factors absent from our in vitro assays may collaborate to regulate Nwk in vivo.

R2.16 Figure 4A and B. The concentration of Nwk and Dap160 are different in A and B, which appear the change the actin assembly curves (compare the red curve 5 in A and curve 4 in B). The authors concluded that neither Nwk, PI(4,5)P2 or NWK^+^PI(4,5)P2 on their own were sufficient to activate WASp above baseline. Was that because of lower Nwk concentration used in B? Why was smaller concentrations of Nwk and Dap160 used in B? Would similar enhancement of actin assembly be observed if 10% PI(4,5)P would be supplemented in the experiments shown in A?

The reviewer is correct: in A, where we were testing the ability of Dap160SH3CD to activate Nwk/Wsp in the absence of membranes, we used 500 nM Nwk and 2 µM Dap160. In B, where we were testing potentiation by membranes, we used 100 nM Nwk and 500 nM Dap160. This was to ensure that we were in a sensitive range to detect further activation by membranes. We chose these concentrations based on our previously published experiments (Stanishneva-Konovalova, Kelly et al. 2016, Figure 4), where we observed minimal release of 100 nM Nwk from autoinhibition by WASp and membranes alone. We would expect to see the same general effect if PIP2 were added to A, but since we might be close to maximal activity for NwK^+^Wsp+Dap160 at these higher concentrations, we may not have been able to effectively detect additional effects of membranes.

R2.17 Table 1 and Figure 5E-F, 3—figure supplement 2D: These seems to be a mismatch between the *Drosophila* genotypes of control and Dap160 RNAi. What is UAS-dcr2/+ which is not present in the control RNAi strain.

Dcr2 is a component of the RNAi-processing machinery, and its overexpression improves knockdown by dsRNA. Dcr2 was included in both genetic backgrounds; the table has been updated to fix this omission.

R2.18 The role of the PAZ proteins in membrane remodelling is addressed with GUV liposomes in vitro, and this message would be stronger if the authors would address the role of these proteins driving membrane remodelling in response to physical cues in in vivo (for example doing EM of NMJ from different *Drosophila* genotypes that show endocytic defects).

While we agree that this would be amazing, the technology does not exist at *Drosophila* synapses in vivo for capturing specific endocytic membrane remodeling events by EM. This is because synaptic boutons are ~5 µm diameter structures filled with hundreds of synaptic vesicles, which obscure each other and surrounding structures, making it very difficult to identify and resolve internalization events. Thin section electron microscopy or even tomography can capture gross steady-state phenotypes such as overall depletion of synaptic vesicles in response to massive depolarization or a train of action potentials, or changes in the steady-state size or spatial distribution of synaptic vesicles. These experiments have previously been done in Dap160 and Nwk mutant synapses (Coyle et al., 2004; Koh et al., 2004, 2007; Marie et al., 2004), and at rest there is no dramatic change in synaptic vesicle or plasma membrane morphology, while upon strong induction of exocytosis, the reformed vesicles form large cisternae, which can resolve over time (Winther et al., 2013, 2015). These results suggest that endocytic phenotypes, especially those that quantitatively affect kinetics or frequency of events, would be impossible to discern by EM. On the other hand, our actin fluorescence microscopy experiments are the first time, to our knowledge, that discrete events associated with endocytic machinery have been visualized and quantified. Applying an approach like correlative light microscopy and tomography or cryotomography to these events would be incredible, but also incredibly technically challenging and difficult to do at sufficient N to quantify mutant phenotypes.

Reviewer #2 (Significance (Required)):The role of actin in presynapses has been widely studied, but the spatio-temporal molecular mechanisms that control the actin dynamics is not well understood. Presynaptic imaging is challenging due to small size of the presynapses and the highly crowded presynaptic environment. This manuscript addresses actin dynamics in vivo in *Drosophila*, which is an impressive feat. The conceptual findings of this study are important for the description of the synaptic function, as well as broader understanding of the periactive zone organization and regulation. The manuscript provides important insights into how synapses auto-regulate their functions that are important for synaptic plasticity and homeostasis. These findings will be interesting for a broad readership and may serve as an opening for further studies related to the physiological implications of the autoregulatory mechanisms.Reviewer's expertise is the fields of cell and molecular biology, and neurobiology, with methodological viewpoint of imaging (conventional and super-resolution microscopy, as well as electron microscopy). The reviewer is not an expert in Drosphila studies or *Drosphila* genetics.